# A numerical method to improve the spatial interpolation of water vapor from numerical weather models: a case study in South and Central America

Fernández Laura I.[1,2], Meza Amalia M.[1,2], Natali M. Paula[1,2], and Bianchi Clara E.[1]

[1]MAGGIA Lab. Fac. de Cs. Astronómicas y Geofísicas. Univ. Nac. de La Plata. Buenos Aires. Argentina.
[2]CONICET, Argentina.

**Correspondence:** Laura I. Fernández (lauraf@fcaglp.unlp.edu.ar)

**Abstract.** Commonly Numerical Weather Models (NWM) users can get the vertically Integrated Water Vapor (IWV) value at a given location from the values at nearby grid points. In this study we used a validated and free available Global Navigation Satellite Systems (GNSS) IWV data set to analyze the very well-known effect of height differences. To this aim, we studied the behavior of 67 GNSS stations in Central and South America with the condition of having a minimum of 5 years of data during the period from 2007 till 2013. The values of IWV from GNSS were compared with the respective values from ERA Interim and MERRA-2 in the same period. Firstly, the total set of stations was compared in order to detect in which cases the geopotential difference between GNSS and NWM deserves a correction. Then, an additive integral correction to the IWV values from ERA Interim was proposed. For the calculation of this correction the multilevel values of specific humidity and temperature given at 37 pressure levels by ERA Interim were used. The performance of the numerical integration method was tested by accurately reproducing the IWV values at each of the grid points surrounding each of the GNSS sites under study. Finally, and considering the $IWV_{GNSS}$ values 6 as a reference, the improvement introduced to the $IWV_{ERAInterim}$ values after adding the corrections is analyzed. In general, the corrections are always recommended but they are not advisable at sea coastal areas or on islands since at least two grid points of the model are usually in the water. In such cases the additive correction could overestimate the IWV.

**Keywords:** 3394 Instruments and techniques; 6904 Atmospheric propagation; 6964 Radio wave propagation.

## 1 Introduction

Water vapor is an abundant natural greenhouse gas of the atmosphere. The knowledge of its variability in time and space is very important to understand the global climate system (Dessler et al., 2008). Most of the regional comparisons of IWV from GNSS are aimed at validating the technique by comparing with radiosonde and microwave radiometers where available. An example of this is the work of Van Malderen et al. (2014) who compared IWV GPS (Global Positioning System) with IWV derived from ground-based sun photometers, radiosondes and satellite-based values from GOME, SCIAMACHY, GOME-2 and AIRS instruments at 28 sites in the northern hemisphere. Because their comparison is oriented to climatology application, they deal

with long-term time series (+ 10 years). The authors asseverate that the mean biases of the GPS with the different instruments vary only between -0.3 and 0.5 $kg\ m^{-2}$ but there are large standard deviations especially for the satellite instruments.

However, some other comparisons examine the $IWV_{GNSS}$ values with respect to the respective estimates from Numerical Weather Models (NWM). If focusing on the application of the current state-of-the-art reanalysis ERA-Interim from the European Centre for Medium-Range Weather Forecasts (ECMWF), both in local and global scale, some recent papers deserve to be mentioned: Heise et al. (2009) used ground pressure data from ECMWF to calculate IWV from 5-minutes Zenith Total Delay (ZTD) at stations without meteorological data available. The authors validate their results with stations with local measurements of pressure and temperature. They also compare IWV from GPS with respect to IWV from ERA-Interim on a global scale. The authors found that IWV from GPS and ECMWF show well agreement on most stations on the global scale except in mountain regions. Moreover they addressed that temporal station pressure interpolation may result in up to 0.5 $kg\ m^{-2}$ IWV uncertainty if a local weather event happened. According to the authors, this phenomenon is observed especially in the tropics and is due to the fact that the ECMWF analysis does not adequately represent the local situation if facing with an increase in the diurnal cycle of surface atmospheric pressure.

Buehler et al. (2012) compare IWV values over Kiruna in the north of Sweden from five different techniques (radiosondes, GPS, ground-based Fourier-Transform InfraRed (FTIR) spectrometer, ground-based microwave radiometer, and satellite-based microwave radiometer) with IWV from ERA-Interim reanalysis. The processed GPS dataset covers a ten-year period from November 1996 to November 2006. The authors found a good overall agreement between IWV from GPS and from ERA-Interim being the mean of differences 0.29 kg m$^{-2}$ and its standard deviation 1.25 kg m$^{-2}$. They also point out that ERA-Interim is drier than the GPS at small IWV values and slightly moister at high IWV values (above 15 kg m$^{-2}$). The authors also consider altitudes limits when comparing measurements from different data sets. They proposed an empirical solution by computing linear regression slopes as a function of the height and corrected all measurements to a common reference altitude of 430 meters. Thus, they established a relative bias of -3.5 % per 100 meters that introduced absolute errors below 0.2 kg m$^{-2}$. Nevertheless they asseverate that the good performance of the method depends on location and probably on season.

Ning et al. (2013) evaluate IWV from GPS in comparison with IWV from ERA-Interim and IWV from the regional Rossby Centre Atmospheric (RCA) climate model at 99 European sites for a 14-year period. Because RCA is not an assimilation model, the standard deviation of RCA-GPS resulted 3 times larger than ERA-GPS. The IWV difference for individual sites varies from -0.21 up to 1.12 kg m$^{-2}$ for ERA-GPS and the corresponding standard deviation is 0.35 kg m$^{-2}$. They investigate the influence of the differences between NWM and GPS in the vertical and horizontal positions. In particular, the authors studied sub-sets of stations with absolute value of height differences smaller than 100 meters. Consequently they do not take into account if there were an over or underestimation by the models. Thus, they found values of the monthly mean IWV differences smaller than 0.5 kg m$^{-2}$. Moreover, the authors also highlight that the models overestimate IWV for sites near the sea.

Bordi et al. (2014) studied global trend patterns of a yearly mean of IWV from the $20^{th}$ century atmosphere model (ERA-20CM) and ERA-Interim, both from ECMWF. The authors highlight a regional dipole pattern of inter-annual climate variability over South America from ERA-Interim data. According to this study, the Andean Amazon basin and Northeast Brazil

are characterized by rising and decreasing water content associated with water vapor convergence (divergence) and upward (downward) mass fluxes, respectively. Besides, the authors also compared IWV from ERA-Interim with the values estimated at 2 GPS stations in Bogotá and Brasilia. Such comparison on monthly timescale made known a systematic bias attributed to a lack of coincidence in the elevation of the GPS stations and the model grid points.

Tsidu et al. (2015) presented a comparison between IWV from a Fourier Transform InfraRed spectrometer (FTIR, at Addis Ababa), GPS, radiosondes, and ERA-Interim over Ethiopia for the period 2007-2011. The study is focused on the characterization of the different error sources affecting the data time series. In particular, from the study of diurnal and seasonal variabilities, the authors addressed differences in the magnitude and sign of the IWV bias between ERA-Interim and GPS. They linked this effect with the sensitivity of the convection model with respect to the topography.

Wang et al. (2015) performed a 12-year comparison of IWV from 3 third generation atmospheric reanalysis models including ERA-Interim, MERRA and the Climate Forecast System Reanalysis (CFSR) on a global scale. IWV values from the reanalysis models were also compared with radiosonde observations in land and Remote Sensing Systems (RSS) on satellites over oceans. The authors asseverate that the main discrepancies of the 3 datasets among them are in Central Africa, Northern South America, and highlands.

In this paper, we investigate the differences between IWV from GNSS by using data products from (Bianchi et al., 2016a) and IWV values given by ERA-Interim and MERRA-2. The comparison was performed taking into account the geopotential differences ($\Delta z$) between each GNSS station and the corresponding values assigned by the models. We proposed an additive numerical correction to the IWV from NWM and the strategy was tested for ERA-Interim re-analysis model. Section 2 describes the different sets of data used in this study. Following, we give the explanation of the methodology and the presentation 20  of the results obtained after applying the proposed correction to the IWV values from ERA-Interim.

## 2  Data

### 2.1  IWV from GNSS

The GNSS data is the main source of information for the spatial and temporal distribution of water vapor. Thus, the main variable considered is the IWV estimated from the delay caused by the troposphere to the GNSS radio signals during its travel 25  from the satellite to the ground receiver. The total delay projected onto the zenith direction (ZTD) is usually split into two contributions: the hydrostatic delay (ZHD, Zenith Hydrostatic Delay) depending merely on the atmospheric pressure and the Zenith Wet Delay (ZWD) depending mainly on the humidity. The $IWV_{GNSS}$ can be obtained from ZWD multiplying it by a function of the mean temperature of the atmosphere.

The reference database of $IWV_{GNSS}$ used in this study comes from a geodetic processing over 136 tracking stations in 30  the American continent located from southern California to Antarctica (see Figure 1), during the 7-year period from January 2007 till December 2013 (Bianchi et al., 2016b). Specifically, the data series of $IWV_{GNSS}$ used here is restricted to those 67 stations with IWV time series spanning more than 5 years (see Figure 1 and Table 1). More details of the steps, models and conventions followed by the geodetic processing to obtain the $IWV_{GNSS}$ values are in Bianchi et al. (2016a).

## 2.2 IWV from NWM

The values of columnar Integrated content of Water Vapor (IWV) as reanalysis products from ERA-Interim (Dee et al., 2011) and MERRA-2 (GMAO, 2015;Bosilovich et al., 2015;Gelaro et al., 2017) were evaluated in this study. The horizontal resolutions are $0.25° \times 0.25°$ for ERA-Interim and $0.625° \times 0.50°$ for MERRA-2. Because ERA-Interim data is given 4 times a
5   day, in order to perform the comparison and even if MERRA-2 gives hourly data, we pick up IWV data from MERRA-2 every 6 hours at 0, 6, 12 and 18 hours of Universal Time (UT). Thus, to be able to carry out the comparison, MERRA-2 was only partially evaluated.

ERA-Interim is the global atmospheric reanalysis produced by the European Centre for Medium-Range Weather Forecasts (ECMWF). It covers the period from 1979 up to today and supersedes the ERA-40 reanalysis. ERA-Interim address some
difficulties of ERA-40 in data assimilation mainly related to the representation of the hydrological cycle, the quality of the stratospheric circulation, and the consistency in terms of reanalyzed geophysical fields (Dee et al., 2011).

MERRA-2 is the successor of The Modern-Era Retrospective Analysis for Research and Applications (MERRA) from NASA's Global Modeling and Assimilation Office (Rienecker et al., 2011). MERRA-2 is improved because it contains less trends and jumps linked to changes in the observing systems than MERRA. Additionally, MERRA-2 assimilates observations
not available to MERRA and reduces bias and imbalances in the water cycle (Gelaro et al., 2017). Moreover, the longitudinal resolution of MERRA-2 data is changed from $0.667°$ in MERRA to $0.625°$ whereas the latitudinal resolution remains unchanged ($0.5°$) (Bosilovich et al., 2015).

To this application we used two different kind of data sets: The 2-D values of the IWV from both re-analysis models along with the correspondent geopotential invariant. We also used three 3-D data sets from ERA-Interim: the air temperature ($T$), the
specific humidity ($q$) and the geopotential ($z$). These variables are given in 37 levels of atmospheric pressure from 1 to 1000 hPa.

We will denominate as $z_i, T_i, p_i$ and $q_i$ to the value of the before mentioned variables at a given $i$ level and in a given instant . This set of data will be used for the calculation of the integral correction that will be developed in the following section.

## 3   Methodology

As mentioned before, even when both reanalysis models give grid values of the vertical integral of the water vapor, the solution provided by each model is linked to its respective geopotential surface invariant. Nevertheless, elevation differences between geopotential from each model grid and computed from GNSS height must be addressed.

In order to compute the geopotential of the GNSS stations ($z_{GNSS}$) we followed the van Dam et al. (2010) algorithm. Because the geodetic coordinates ($\phi, \lambda, h$) of the GNSS site are known, we obtained the orthometric height ($H$) at each GNSS
station by correcting the ellipsoidal height with the EGM08 model (Pavlis et al., 2012). Thus, the geopotential is (van Dam et al., 2010)

$$z_{GNSS} = \frac{g_s(\phi)\,C(\phi)\,H}{(C(\phi) + H)} \qquad (1)$$

where the radius of the ellipsoid at geodetic latitude $\phi$ is,

$$C(\phi) = \left( \frac{cos^2(\phi)}{a^2} + \frac{sin^2(\phi)}{b^2} \right)^{-1/2} \tag{2}$$

with $a = 6378137m.$ and $b = 6356752.3142m.$ being the semimajor and semiminor axis of the WGS84 ellipsoid, respectively (Hofmann-Wellenhof and Moritz, 2006). Moreover, the value of the gravity on the ellipsoid at geodetic latitude $\phi$ can be
written as (van Dam et al., 2010).

$$g_s(\phi) = g_E \frac{1 + k_s \, sin^2(\phi)}{\sqrt{1 - e^2 \, sin^2(\phi)}} \tag{3}$$

where $e^2 = 0.00669437999014$ is the first eccentricity squared of the WGS84 ellipsoid and $g_E = 9.7803253359m \, s^{-2}$ is the normal gravity at the Equator (Hofmann-Wellenhof and Moritz, 2006) and $k_s = 0.001931853$ (van Dam et al., 2010).

For a given GNSS station, the respective geopotential from each of the 2 reanalysis models resulted from a bi-linear in-
10 terpolation of the respective invariant static geopotential at the 4 grid points around the GNSS site, referred to as $z_{NWM}^k$
$(k = 1, 2, 3, 4)$. Because the points of the NWM grid surrounding the GNSS station have different geopotential values and those values are in turn different from $z_{GNSS}$, we propose to correct on each of these 4 points. Thus, if $\Delta z^k$ refers to the difference between $z_{GNSS}$ and $z_{NWM}^k$,

$$\Delta z^k = z_{GNSS} - z_{NWM}^k, \quad k = 1, 2, 3, 4. \tag{4}$$

where NWM corresponds to ERA-Interim or MERRA-2.

We will then propose a correction procedure that, compensating $\Delta z^k$ in each of the 4 grid points, corrects the values of $IWV_{NWM}$. After "moving" the grid points to $z_{GNSS}$, a bi-linear interpolation is performed to obtained the corrected value of IWV at the location of the GNSS site. In brief, this procedure is equivalent to lifting (or descending as appropriate) each of the grid points in order to build up a plane at $z_{GNSS}$.

Prior to the correction, we analyze the performance of ERA-Interim and MERRA-2 with respect to GNSS. Thus, although $IWV_{GNSS}$ is produced every half hour and $IWV_{MERRA-2}$ is available hourly, we consider only the epochs when ERA Interim data is available to perform the comparison at 0, 6, 12 and 18 UT. Table 2 shows the mean values of IWV from GNSS ($\overline{IWV}_{GNSS}$) during the period 2007-2013 and its standard deviation for all the stations. We assume that the static geopotential from the NWM at each GNSS site ($z_{NWM}$) is obtained from a bi-linear interpolation of the static geopotential at the 4 grid
points surrounding it ($z_{NWM}^k$). Thus, Table 2 shows the geopotential difference ($\Delta z = z_{GNSS} - z_{NWM}$) in addition to the respective differences of the mean values ($\Delta \overline{IWV} = \overline{IWV}_{GNSS} - \overline{IWV}_{NWM}$), where NWM indicates ERA Interim and MERRA-2. All the averages are computed over the period 2007-2013.

In general when regarding at Table 2, we can observe that the best agreements between the average IWV values from GNSS and the corresponding average from the models are where the $\Delta z$ are small (for example: CONZ, VITH, SMAR, LPGS,
MAPA, SCUB among others). In other words, in general the NWMs represent very well the IWV values ($\Delta \overline{IWV} < 1.5$ $kg \, m^{-2}$) if $|\Delta z|$ is small. That means, the geopotential difference is in the order of 500 $m^2 \, s^{-2}$ at most.

On the other hand, the difference of the model representation of the IWV with respect to GNSS grows as the height differences ($\Delta z$) become larger and this is true for all values of $\overline{IWV}_{GNSS}$. As an example we can mention the cases of SANT ($\overline{IWV}_{GNSS} \sim 12\,kg\,m^{-2}$, $|\Delta z| > 10000\,m^2\,s^{-2}$) and CUCU ($\overline{IWV}_{GNSS} \sim 43\,kg\,m^{-2}$, $|\Delta z| \sim 8000\,m^2\,s^{-2}$).

However, other than these cases that can be considered critical, the differences are also important in those sites with moderate
$|\Delta z|$ (larger that $500\,m^2\,s^{-2}$) and $\overline{IWV}_{GNSS} > 20kg^{-2}$ (CEFE, BRAZ, RIOD, GUAT).

Note that some MERRA-2 differences values could be a little bigger than ERA Interim ones and this would be expected because of the coarser grid. However, this is not a general rule and some stations are in fact better represented by MERRA-2 with $|\Delta z|$ and $\left|\Delta\overline{IWV}\right|$ smaller than ERA Interim even if they are located in highlands (see SANT, COPO).

Figure 2 (up) shows the the mean IWV values from GNSS ($\overline{IWV}_{GNSS}$) as a function of geopotential differences ($\Delta z$).
Results for MERRA-2 are on the left and the same for ERA-Interim on the right.

It is assumed that these different values of $\Delta\overline{IWV}$ are due to the geopotential difference. Therefore, they commonly carry the opposite sign to the $\Delta z$ as we expect. Effectively, $\Delta\overline{IWV}$ is nothing but the difference between mean values from GNSS and NWM, a negative value for negative $\Delta z$ indicates an overestimation by the reanalysis model and on the contrary, the underestimation by the model is shown with red dots where $\Delta z$ is positive.

However the figure 2 (up) shows that there are some cases where $\Delta\overline{IWV}$ have the same sign as $\Delta z$ evidencing that the reanalysis models can overvalue or undervalue the value of $\overline{IWV}$ and this should not be due to $\Delta z$. This effect can be seen within a rectangle whose size are $\Delta z = \pm\,2000\,m^2s^{-2}$ and $\Delta\overline{IWV} = \pm\,2\,kg\,m^2$. Such a cloud of points represent about $21\%$ of the total number of stations for ERA-Interim and a $27\%$ for MERRA-2. Figure 2 (down) shows the spatial distribution of $\Delta\overline{IWV}$ for MERRA-2 (left) and ERA Interim (right). The red dots indicate positive differences greater than $2.5\,kg\,m^2$ and the intermediate differences between 0 and $2.5\,kg\,m^2$ are in a red degrade. Similarly, the blue dots show negative differences less than $-2.5\,kg\,m^2$ and the differences between $-2.5$ and $0\,kg\,m^2$ are in a blue degrade.

If we take as reference the RNNA station in both maps (see station number 51 in Figure 1), and advancing towards the south along the Atlantic coast, the behavior of both models results similar. Both reanalysis are dryer than GNSS and this same effect is seen in the southern mountainous areas. However, going along the Atlantic coast from RNNA to the north and up to the Amazon river we can see different behaviours of the reanalyses, while ERA Interim continues underestimating $IWV_{GNSS}$, MERRA-2 shows to be wetter than GNSS. The overall agreement between GNSS and MERRA-2 is $-0.39 \pm 2.77$ and between GNSS and ERA Interim is $0.13 \pm 2.52$. Being these values the result of an average over all the $\Delta\overline{IWV}$ differences.

These findings show that MERRA-2 resulted wetter than GNSS while ERA Interim is slightly dryer than GNSS in Central and South America. This is in agreement with the findings of Buehler et al. (2012), who found out that the mean value of the differences between IWV from GPS and ERA is $0.28 \pm 1.25$ for a high latitude location in Sweden, revealing also an underestimation of the reanalysis model.

Finally, the correlation coefficients between $\overline{IWV}_{GNSS}$ values and the respective ones for both NWM, are higher than 0.95 in most of the GNSS stations (not shown).

### 3.1 Computation of the integral correction

Following we proceed to calculate a correction in order to provide a better estimation of the $IWV_{NWM}$ at the GNSS site. We start by correcting each of the grid points around the station prior to apply a bi-linear interpolation. The correction will be computed only for one of the two re-analysis models tested. We have chosen ERA Interim over MERRA 2 for the calculation and testing of these corrections not only because ERA Interim has a thinner grid, but also considering the results of Zhu (2014). Effectively, Zhu (2014) compared several reanalysis projects with independent sounding observations recorded in the Eastern Himalayas during June 2010. Among all the reanalysis models ERA-Interim and MERRA, the predecessor of MERRA 2 were included. The authors analyze temperature, specific humidity, u-wind, and v-wind between 100 hPa and 650 hPa. The authors found that ERA-Interim showed the best performance for all variables including specific humidity, the key variable to produce the integrated water vapor.

Thus, we used air temperature ($T_i$) and specific humidity ($q_i$) on 37 atmospheric pressure levels from ERA-Interim data to compute the proposed correction.

Recall that the index $k$ refers to the grid point surrounding the GNSS site while the index $i$ refers to the atmospheric pressure level. As we mentioned before, the GNSS geopotential ($z_{GNSS}$) is set as a reference, and the value of the geopotential from ERA-Interim ($z_{ERAInterim}^k$) at each of the 4 grid points surrounding the GNSS site are generally not the same and could differ up to two orders of magnitude. Commonly, neither $z_{GNSS}$ nor the geopotential at any of the 4 grid points matches the geopotential of the nearby pressure level. Therefore, the values of all parameters in the adjacent levels must be used to interpolate (or extrapolate) pressure, temperature and specific humidity in the unknown geopotential ($z_{GNSS}$ and $z_{ERAInterim}^k$).

Thus, the expression of the pressure at an unknown geopotential $z_j$, where $j$ can be any of the unknowns, with respect to a given reference data level ($z_0$) at $i = 0$ is (van Dam et al., 2010)

$$p(z_j) = p_0 \left( \frac{T_0 - \lambda \, \delta z}{T_0} \right)^{g_0/R\lambda} \tag{5}$$

where $T_0$ and $p_0$ refer to the temperature and pressure values at a reference level $z_0$, $R = 287.04 \; J \, kg^{-1} \, K$ is the gas constant and $\lambda = 0.006499 \; K \, m^{-1}$ is the lapse rate of the temperature, and $\delta z$ is the geopotential difference between $z_i$ and the reference level $z_0$. Notice that $\delta z$ is different of $\Delta z$, where the $\Delta z$ refers to the difference between $z_{GNSS}$ and $z_{NWM}$. The numerator of Eq. (5) is the temperature estimated at the desired geopotential $z_j$ assuming that the temperature decreases with altitude according to $\lambda$. This expression is used to compute $p$ both in $z_{GNSS}$ and in the four grid points of the model ($z_{ERAInterim}^k$). Finally, the specific humidity ($q$) is also estimated at the desired $z_j$ by a linear interpolation (extrapolation) from data at the adjacent layers.

After knowing $p, T$ and $q$ at each geopotential, $z_{GNSS}$ and the 4 grid points of $z_{ERAInterim}^k$, we can estimate the necessary corrections to the grid points. Such additive corrections to the IWV values at the grid points are equivalent to move the static geopotential of the grid to the $z_{GNSS}$. Then, the corrected $IWV_{NWM}$ is obtained at the GNSS site by a bi-linear interpolation of the 4 corrected values.

Each value of IWV provided by ERA-Interim is the result of the numerical integration of the expression (Berrisford et al., 2011).

$$IWV_{ERAInterim} = \frac{1}{g_0} \int_{p_1}^{p_s} q(p) \, dp \tag{6}$$

where $g_0$ is the standard acceleration of the gravity at mean sea level, $q(p)$ is the specific humidity of the air at the pressure
level $p$ and the integral is calculated from the first level ($p_1$) up to the model surface level ($p_s$), i.e. up to the static geopotential that corresponds to the station.

Thus, the proposed correction can generally be written by:

$$\Delta IWV = \frac{1}{g_0} \sum_{j=A}^{B} \frac{q_{j+1} + q_j}{2} (p_{j+1} - p_j) (-1)^n \begin{cases} n = 1 & if \quad p_{GNSS} < p_{NWM} \\ n = 2 & if \quad p_{GNSS} > p_{NWM} \end{cases} \tag{7}$$

where the NWM is ERA Interim, A corresponds to the highest $z$ ($z_{GNSS}$ or $z_{NWM}^k$) and B the lowest $z$. Note that the values
of $q$ and $p$ in equation (7) can be computed as explained before (equation 5). Thus, $q_j$ and $p_j$ are $q$ and $p$ at $z_j$ and $q_{j+1}$ and $p_{j+1}$ are $q$ and $p$ at $z_{j+1}$, respectively. The values of $p$ grow downwards resulting $p_1$ = 1 hPa. and $p_{37}$= 1000 hPa. Assuming that the integral of the water vapor is computed from topside and downwards, if the height of a given point from a model is located lower than the position of the receiver, the model integrates a larger column of water vapor and the opposite if the geopotential value from model is larger than the geopotential of the GNSS receiver. Hence, this quantity has to be additive if
$z_{GNSS} < z_{NWM}^k$ or subtractive if opposite and the sign is determined by $n$ (see Eq. (7)).

In a given instant, we know the geopotential of the GNSS station and the static geopotential assigned by the NWM to the 4 grid points surrounding it ($z_{GNSS}$ and $z_{NWM}^k$, k = 1,2,3,4). We also know the geopotential at 37 pressure levels ($z_i$) from 1 hPa till 1000 hPa, as well as specific humidity ($q$) and temperature ($T$) at these levels. We should consider that at any time the pressure value of each level is constant but this is not necessarily the case for geopotential height.

Figure 3 illustrates the application of the correction to an example. We take just 1 of the 4 grid points and lets suppose that both unknowns ($z_{NWM}^k$ and $z_{GNSS}$) are located between the levels 27 (750 hPa) and 28 (775 hPa). Thus, we could use the available data at levels 27 and 28 along with Eq. (5) and the before mentioned considerations to estimate $p, t$ and $q$ at $z_{NWM}^k$ and $z_{GNSS}$. Finally, $\Delta IWV$ is computed by means of Eq. (7) for this example.

## 4    Results

Before analyzing the results of the correction process explained in the previous section, we will present a validation of the numerical integration method used. To this end, we calculate the values $IWV_{ERAInterim}$ at each grid point by using the numerical integral of the equation 6. The integration limits range from 1 hPa to the static geopotential value assigned by the model to the point ($z_{ERAInterim}^i$). Table 3 shows the results obtained with this procedure. In each grid point the mean value of the differences ($IWV_{ERAInterim}$ data - $IWV_{ERAInterim}$ calculated) is presented. Standard deviations are also shown. It can
be seen that in general the resulting values are very close to zero.

In order to evaluate the improvements introduced by the correction, figure 4 shows the $\Delta\overline{IWV}$ as a function of $\Delta z$ after applying the proposed integral correction to ERA-Interim data. At first glance we can see that, regardless of whether $\Delta z$ is positive or negative, the differences $\left|\Delta\overline{IWV}\right|$ decrease to $2\ kg\ m^{-2}$ for all the stations except in 3 cases where they barely exceed that value. Moreover, the correlation between $\Delta\overline{IWV}$ and the geopotential difference decreases to 0.13 as expected.

In addition, if we focus on the plot area of figure 4 limited for $\Delta z = \pm\ 1500\ m^2\ s^{-2}$ and $\Delta\overline{IWV} = 1.5\ kg\ m^{-2}$ (zoom not shown), we can see that most of the stations (94 %) are in there. This shows that the proposed correction decreases $\Delta\overline{IWV}$ even for low stations (small z) which generally have the smallest values of $\Delta z$.

The performance of the proposed correction can also be seen in Figure 5. The plots are arranged in two columns where the left column shows stations with positive $\Delta z$, it means that GNSS station is higher to the location assigned by ERA-Interim. Accordingly, the model integrates a thicker layer of atmosphere and thus $IWV_{ERAInterim}$ values resulted larger than ones from $IWV_{GNSS}$. The opposite ($\Delta z$ is negative) is represented by the sites at the right column. Moreover, the differences in $\Delta z$ are presented increasing from top to bottom in each column.

We can see that the most important corrections are at BOGT in Bogotá, Colombia, and SANT in Santiago de Chile, Chile. In these examples the differences ($IWV_{GNSS} - IWV_{ERAInterim}$), which can reach up to $7\ kg\ m^{-2}$, are significantly reduced.

However the application of this correction, in some cases, should be precautionary. Effectively, sometimes different shortcomings of the model overlap the height problem and therefore the proposed correction could not work. As an example of this we can mention the case of coastal and/or insular stations where 2 or more grid points will be in the ocean. In all these cases the value of IWV calculated from the bi-linear interpolation will be overvalued. Let's analyze in detail the case of stations near the seashore (for example PARC in Punta Arenas, Chile) where 2 of the 4 grid points are in the ocean (see Figure 6). Also $\Delta z$ = -1271.86 in PARC indicating that the geopotential from ERA Interim is larger than the GNSS geopotential and therefore the proposed correction will be additive. Besides this result, the $IWV_{ERA-Interim}$ resulted over-estimated by applying a bi-linear interpolation that uses data points in the ocean. In conclusion, the value ($IWV_{ERA-Interim} + correction$) will overestimate $IWV_{GNSS}$. Thus, this is an example where applying the suggested correction may worsen the results. The same situation is presented in RIO2 at the Argentinean Atlantic coast.

## 5   Discussion and Conclusions

The effect of different heights when comparing results from several data sources not only affects the determination of IWV but also other parameters. For instance, Gao et al. (2012) studied the height corrections for the ERA-Interim 2m-temperature data at the Central Alps and they also found large biases that must be corrected in mountainous areas. Some other authors, also studied the tropospheric refraction effects on space geodetic techniques by considering this effect. For example, Teke et al. (2013) performed an inter-technique comparison of ZTD in the framework of 4 continuous Very Long Baseline Interferometry (VLBI) campaigns also including NWM and taking into account the effect of the height differences.

The NWM users commonly utilize the IWV values on a grid and calculate with them the IWV value at the desired place by using some interpolation method. In this work, taking the values of $IWV_{GNSS}$ as reference, we show that there are cases

where the IWV values obtained from the NWM have differences of several $kg\,m^{-2}$ and these discrepancies are mainly due to the difference in geopotentials.

We analyzed the discrepancies between the vertically Integrated Water Vapor values provided by two re-analysis models (ERA-Interim and MERRA-2) with respect to the $IWV_{GNSS}$ values taken as a reference in the South and Central American continent for the period 2007-2013. The results of this comparison allow us to ensure that MERRA-2 resulted wetter than GNSS while ERA Interim is slightly dryer. In addition, when geopotential differences are moderate or large ($|\Delta z| > 500$ $m^2\,s^{-2}$) and $\overline{IWV}_{GNSS} > 20\,kg\,m^{-2}$, the discrepancies ($\Delta\overline{IWV}$) are greater than $2\,kg\,m^{-2}$ at about 22% of the stations for both models.

Several authors reported problems related to the elevation correction for data from the reanalysis models. The artificial bias in IWV introduced by this altitude difference was previously reported by Bock et al. (2007); Heise et al. (2009); Van Malderen et al. (2014); Bordi et al. (2014) and Bianchi et al. (2016a).

Heise et al. (2009) derived IWV from global GNSS ZTD at almost 300 sites by using ground pressure and temperature values from ECMWF and then compared IWV from GNSS and ECMWF. Similar to our work, they found large discrepancies in mountain regions due to the difference in altitudes that caused errors in the meteorological values estimations. Moreover, the analysis performed in our work is also in agreement with Bordi et al. (2014) who compared between GNSS and ERA-Interim IWV values but on monthly time scale for the period 2002-2012. In this case the authors found significant biases of $6.4\,kg\,m^{-2}$ in BOGT and $2.5\,kg\,m^{-2}$ in BRAZ and they related them to the different elevations between the correspondent GNSS site and the grid points of the model. Thereby, the values of Bordi et al. (2014) are comparable with the correspondent $|\Delta\overline{IWV}|$ values estimated by our study ($6.87\,kg\,m^{-2}$ in BOGT and $2.03\,kg\,m^{-2}$ in BRAZ, see Table 2).

In this work we proposed an integral correction that compensates on IWV the effect of the geopotential difference between GNSS and the interpolated grid points in the reanalysis model. The results were tested with the respective ones from ERA-Interim. The correction is computed as the numerical integration of the specific humidity where the integral limit is a pressure difference at $\Delta z$ (see Eqs. 7). Taking into account that prior to the correction, the 67% of the stations have $|\Delta\overline{IWV}| < 1.5$ $kg\,m^{-2}$ for ERA-Interim, the application of the numerical correction improves the results and thus the percentage of stations below $1.5\,kg\,m^{-2}$ increased to 94%.

Nevertheless, the application of this correction is not advisable at coasts and insular stations of South and Central America because the overvaluation of the model near the coast is overlapped to the height problem. These results are in agreement with Ning et al. (2013), who also compared IWV from GNSS with values from two NWM (ERA-Interim and Rossby Centre Atmospheric climate model, RCA) in Europe for 14 years. The authors also found that models give IWV values larger than GNSS at the seasides or coasts where the tile of the model includes more than 60% of water.

For this reason, the corrections we propose are always recommended but they are not advisable at sea coastal areas or on islands since at least two grid points of the model are usually in the water.

*Author contributions.* L.I. Fernández led the study and contributed to data collection, analysis, and interpretation of the results; A.M. Meza and M.P. Natali co-wrote the paper. They also contributed to the statistical analysis and the interpretation of the results. C. E. Bianchi contributed to data collection. All authors read and approved the final manuscript.

*Competing interests.* The authors declare that they have no conflict of interest.

5   *Acknowledgements.* We would like to thank the two anonymous reviewers for their valuable comments that highly improved this manuscript. This research was supported by the National Scientific and Technical Council of Argentina (CONICET) PIP 112-201201-00292 and La Plata National University (UNLP) project 11G/142. We would also like to thank the people, organizations and agencies responsible to collect, compute, maintain and openly provide the observations and the products employed in this work: The European Centre for Medium-Range Weather Forecasts (ECMWF) for providing the ERA-Interim reanalysis data (http://apps.ecmwf.int/datasets/). and the Global Mod-10   eling and Assimilation Office (GMAO) from National Aeronautics and Space Administration (NASA, USA) for providing MERRA-2 data (https://gmao.gsfc.nasa.gov/reanalysis/MERRA-2/).

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

Table 1: Geopotential values at the selected GNSS stations. Values of $z_{ERA-Interim}$ and $z_{MERRA-2}$ come from a bi-linear interpolation of the 4 gridded values of $z$ around the GNSS site.

| | Station name | Longitude [°] | Latitude [°] | Geopotential (z) $[m^2 s^{-2}]$ | | |
| | | | | GNSS | ERA-Interim | MERRA-2 |
|---|---|---|---|---|---|---|
| 1 | ACYA | -99.9030 | 16.8380 | 44.4 | 4173.4 | 1640.1 |
| 2 | AREQ | -71.4928 | -16.4655 | 24017.2 | 26377.1 | 37128.0 |
| 3 | AUTF | -68.3036 | -54.8395 | 575.9 | 2263.5 | 3335.3 |
| 4 | AZUL | -59.8813 | -36.7670 | 1385.7 | 1363.2 | 1348.8 |
| 5 | BELE | -48.4626 | -1.4088 | 336.5 | 172.7 | 127.7 |
| 6 | BOAV | -60.7011 | 2.8452 | 831.69 | 1188.44 | 1130.29 |
| 7 | BOGT | -74.0809 | 4.6401 | 25041.7 | 18524.5 | 19392.8 |
| 8 | BRAZ | -47.8779 | -15.9475 | 10969.8 | 9439.9 | 8995.9 |
| 9 | BRFT | -38.4255 | -3.8774 | 299.6 | 678.2 | 267.1 |
| 10 | BRMU | -64.6963 | 32.3704 | 204.1 | 3.4 | 0.6 |
| 11 | BYSP | -66.1612 | 18.4078 | 915.6 | 550.7 | 1651.3 |
| 12 | CEFE | -40.3195 | -20.3108 | 212.3 | 1761.0 | 2224.2 |
| 13 | CHET | -88.2992 | 18.4953 | 96.6 | 361.0 | 184.3 |
| 14 | CHPI | -44.9852 | -22.6871 | 6087.9 | 8681.9 | 8612.8 |
| 15 | CONZ | -73.0255 | -36.8438 | 1571.6 | 1155.3 | 855.0 |
| 16 | COPO | -70.3382 | -27.3845 | 4392.4 | 12005.2 | 10847.6 |
| 17 | CRO1 | -64.5843 | 17.7569 | 114.9 | -14.7 | 10.9 |
| 18 | CUCU | -72.4879 | 7.8985 | 3049.1 | 11058.5 | 15115.1 |
| 19 | CUIB | -56.0699 | -15.5553 | 2306.0 | 2816.2 | 1992.9 |
| 20 | EBYP | -55.8922 | -27.3689 | 1261.2 | 1279.0 | 1482.4 |
| 21 | FALK | -57.8741 | -51.6937 | 379.4 | 71.2 | 142.7 |
| 22 | GUAT | -90.5202 | 14.5904 | 14879.8 | 10900.6 | 13573.9 |
| 23 | IGM1 | -58.4393 | -34.5722 | 340.4 | 179.5 | 188.9 |
| 24 | ISPA | -109.3444 | -27.1250 | 1140.5 | 12.2 | 14.7 |
| 25 | LPAZ | -110.3194 | 24.1388 | 255.9 | 1022.7 | 941.9 |
| 26 | LPGS | -57.9323 | -34.9067 | 136.5 | 152.3 | 113.8 |
| 27 | MABA | -49.1223 | -5.3624 | 1012.8 | 1211.5 | 1384.4 |
| 28 | MANA | -86.2490 | 12.1489 | 651.3 | 1754.7 | 2208.4 |
| 29 | MAPA | -51.0973 | 0.0467 | 195.7 | 257.1 | 375.5 |

Table 1: Geopotential values at the selected GNSS stations. Values of $z_{ERA-Interim}$ and $z_{MERRA-2}$ come from a bi-linear interpolation of the 4 gridded values of $z$ around the GNSS site.

| | Station name | Longitude [°] | Latitude [°] | Geopotential (z) [$m^2 s^{-2}$] | | |
| --- | --- | --- | --- | --- | --- | --- |
| | | | | GNSS | ERA-Interim | MERRA-2 |
| 30 | MARA | -71.6244 | 10.6740 | 419.8 | 713.9 | 432.6 |
| 31 | MDO1 | -104.0150 | 30.6805 | 19873.8 | 12481.3 | 12736.7 |
| 32 | MERI | -89.6203 | 20.9800 | 209.1 | 137.5 | 263.1 |
| 33 | MGBH | -43.9249 | -19.9419 | 9618.6 | 8782.9 | 8625.4 |
| 34 | MSCG | -54.5407 | -20.4409 | 6615.3 | 4368.7 | 4816.4 |
| 35 | MZAC | -68.8756 | -32.8952 | 8208.8 | 15884.1 | 13487.9 |
| 36 | NAUS | -60.0550 | -3.0229 | 1036.9 | 462.9 | 314.5 |
| 37 | OHI2 | -57.9013 | -63.3211 | 92.4 | 1206.9 | 1015.0 |
| 38 | ONRJ | -43.2243 | -22.8957 | 405.1 | 1879.1 | 3463.2 |
| 39 | PALM | -64.0511 | -64.7751 | 138.0 | 1968.4 | 1953.5 |
| 40 | PARC | -70.8799 | -53.1370 | 119.6 | 1391.5 | 1077.4 |
| 41 | PBCG | -35.9071 | -7.2137 | 5276.8 | 3525.6 | 3735.4 |
| 42 | PEPE | -40.5061 | -9.3844 | 3749.1 | 4542.5 | 4201.2 |
| 43 | POAL | -51.1198 | -30.0740 | 703.2 | 1251.9 | 272.3 |
| 44 | POLI | -46.7303 | -23.5556 | 7196.9 | 6505.1 | 4637.3 |
| 45 | POVE | -63.8963 | -8.7093 | 1055.3 | 954.4 | 960.0 |
| 46 | PPTE | -51.4085 | -22.1199 | 4276.3 | 3724.0 | 3862.4 |
| 47 | RECF | -34.9515 | -8.0510 | 252.1 | 936.5 | 419.7 |
| 48 | RIO2 | -67.7511 | -53.7855 | 190.9 | 1005.9 | 465.5 |
| 49 | RIOB | -67.8028 | -9.9655 | 1448.4 | 1821.4 | 1706.4 |
| 50 | RIOD | -43.3063 | -22.8178 | 139.3 | 2320.9 | 2974.6 |
| 51 | RNNA | -35.2077 | -5.8361 | 498.8 | 512.2 | 355.4 |
| 52 | SALU | -44.2125 | -2.5935 | 433.9 | 121.0 | 279.5 |
| 53 | SANT | -70.6686 | -33.1503 | 6817.3 | 17026.0 | 11607.4 |
| 54 | SAVO | -38.4323 | -12.9392 | 855.5 | 412.2 | 821.1 |
| 55 | SCUB | -75.7623 | 20.0121 | 436.4 | 1349.9 | 1839.3 |
| 56 | SMAR | -53.7166 | -29.7189 | 1015.4 | 1997.8 | 1877.5 |
| 57 | SSA1 | -38.5165 | -12.9752 | 87.6 | 458.1 | 964.7 |
| 58 | SSIA | -89.1166 | 13.6971 | 6131.3 | 4299.0 | 5206.1 |

Table 1: Geopotential values at the selected GNSS stations. Values of $z_{ERA-Interim}$ and $z_{MERRA-2}$ come from a bi-linear interpolation of the 4 gridded values of $z$ around the GNSS site.

| | Station name | Longitude [°] | Latitude [°] | Geopotential (z) [$m^2 s^{-2}$] | | |
| | | | | GNSS | ERA-Interim | MERRA-2 |
|---|---|---|---|---|---|---|
| 59 | TOPL | -48.3307 | -10.1711 | 2691.8 | 2752.2 | 3404.1 |
| 60 | TUCU | -65.2304 | -26.8433 | 4475.0 | 9038.4 | 11323.5 |
| 61 | UBER | -48.3170 | -18.8895 | 78698.0 | 7229.8 | 7268.0 |
| 62 | UCOR | -64.1935 | -31.4350 | 4289.4 | 6202.2 | 7133.3 |
| 63 | UFPR | -49.2310 | -25.4484 | 9041.0 | 6861.5 | 7676.6 |
| 64 | UNRO | -60.6284 | -32.9594 | 488.8 | 406.2 | 293.8 |
| 65 | UNSA | -65.4076 | -24.7275 | 12007.0 | 19659.8 | 19240.0 |
| 66 | VESL | -2.8418 | -71.6738 | 8362.1 | 5632.2 | 7588.8 |
| 67 | VITH | -64.9692 | 18.3433 | 479.4 | 74.0 | 36.6 |

Table 2: Differences of the mean values of IWV ($\Delta \overline{IWV}$ in $[kg\,m^{-2}]$) between GNSS and the NWM for the period 2007-2013 at 67 stations located in South America and Central America. The mean value ($\overline{IWV}$) from GNSS at each site is also given and SD refers to the standard deviation. $\Delta z = z_{GNSS} - z_{NWM}$ refers to the difference in the geopotential $[m^2 s^{-2}]$ at each GNSS station.

| | Name | GNSS | | ERA-Interim | | MERRA-2 | |
| | | $\overline{IWV}$ | SD | $\Delta z$ | $\Delta \overline{IWV}$ | $\Delta z$ | $\Delta \overline{IWV}$ |
| | | $[kg\,m^{-2}]$ | $[kg\,m^{-2}]$ | $[m^2 s^{-2}]$ | $[kg\,m^{-2}]$ | $[m^2 s^{-2}]$ | $[kg\,m^{-2}]$ |
|---|---|---|---|---|---|---|---|
| 1 | ACYA | 41.70 | 11.72 | -4129.0 | 4.88 | -1595.7 | 0.92 |
| 2 | AREQ | 11.03 | 6.59 | -2360.0 | 0.33 | -13110.8 | 2.27 |
| 3 | AUTF | 10.48 | 3.71 | -1687.5 | 0.52 | -2759.4 | 0.80 |
| 4 | AZUL | 16.97 | 8.01 | 22.5 | -1.10 | 36.9 | -1.10 |
| 5 | BELE | 49.59 | 6.57 | 163.7 | 0.39 | 208.8 | -1.54 |
| 6 | BOAV | 50.20 | 5.62 | -356.8 | 1.50 | -298.6 | -1.45 |
| 7 | BOGT | 19.54 | 3.14 | 6517.2 | -6.87 | 5648.9 | -8.52 |
| 8 | BRAZ | 26.52 | 9.77 | 1529.9 | -2.03 | 1973.8 | -3.54 |
| 9 | BRFT | 42.26 | 8.05 | -378.6 | 1.10 | 32.5 | -0.49 |
| 10 | BRMU | 29.69 | 11.61 | 200.8 | -0.30 | 203.6 | -0.89 |
| 11 | BYSP | 39.10 | 8.74 | 364.8 | -0.79 | -735.7 | 0.41 |
| 12 | CEFE | 37.60 | 10.75 | -1548.6 | 2.64 | -2011.9 | 2.27 |
| 13 | CHET | 41.82 | 10.45 | -264.4 | 0.69 | -87.7 | -0.93 |
| 14 | CHPI | 29.71 | 10.21 | -2594.0 | 2.03 | -2524.8 | 1.31 |
| 15 | CONZ | 14.16 | 5.32 | 416.3 | 0.09 | 716.6 | -0.35 |
| 16 | COPO | 11.91 | 5.27 | -7612.9 | 3.18 | -6455.2 | 2.49 |
| 17 | CRO1 | 38.53 | 8.76 | 129.6 | -0.79 | 104.0 | -0.98 |
| 18 | CUCU | 43.08 | 5.59 | -8009.5 | 10.53 | -12066.1 | 10.35 |
| 19 | CUIB | 40.98 | 11.98 | -510.2 | 0.82 | 313.1 | -0.25 |
| 20 | EBYP | 28.69 | 12.91 | -17.8 | -0.68 | -221.2 | -1.15 |
| 21 | FALK | 10.91 | 3.95 | 308.2 | -0.41 | 236.7 | -0.68 |
| 22 | GUAT | 22.88 | 7.29 | 3979.2 | -6.75 | 1305.9 | -5.86 |
| 23 | IGM1 | 19.86 | 9.41 | 160.9 | -0.93 | 151.6 | -0.53 |
| 24 | ISPA | 26.26 | 7.40 | 1128.2 | 0.61 | 1125.7 | 0.16 |
| 25 | LPAZ | 25.53 | 15.27 | -766.8 | 0.47 | -686.0 | 0.02 |
| 26 | LPGS | 19.63 | 9.37 | -15.7 | -0.66 | 22.7 | -0.91 |
| 27 | MABA | 46.95 | 7.92 | -198.7 | 0.37 | -371.6 | -1.31 |

Table 2: Differences of the mean values of IWV ($\Delta\overline{IWV}$ in [$kg\,m^{-2}$]) between GNSS and the NWM for the period 2007-2013 at 67 stations located in South America and Central America. The mean value ($\overline{IWV}$) from GNSS at each site is also given and SD refers to the standard deviation. $\Delta z = z_{GNSS} - z_{NWM}$ refers to the difference in the geopotential [$m^2s^{-2}$] at each GNSS station.

| | Name | GNSS | | ERA-Interim | | MERRA-2 | |
| | | $\overline{IWV}$ | SD | $\Delta z$ | $\Delta\overline{IWV}$ | $\Delta z$ | $\Delta\overline{IWV}$ |
| | | [$kg\,m^{-2}$] | [$kg\,m^{-2}$] | [$m^2s^{-2}$] | [$kg\,m^{-2}$] | [$m^2s^{-2}$] | [$kg\,m^{-2}$] |
| 28 | MANA | 44.93 | 9.57 | -1103.5 | 2.27 | -1557.1 | 2.38 |
| 29 | MAPA | 49.92 | 6.67 | -61.3 | 0.25 | -179.8 | -0.51 |
| 30 | MARA | 47.96 | 8.04 | -294.1 | 1.43 | -12.8 | -0.74 |
| 31 | MDO1 | 10.15 | 7.49 | 7392.6 | -5.34 | 7137.1 | -5.39 |
| 32 | MERI | 38.85 | 10.79 | 71.6 | 0.13 | -54.0 | -0.58 |
| 33 | MGBH | 26.75 | 9.88 | 835.9 | -1.00 | 993.2 | -2.63 |
| 34 | MSCG | 31.75 | 10.81 | 2246.6 | -2.74 | 1798.9 | -3.37 |
| 35 | MZAC | 15.25 | 7.33 | -7675.3 | 1.21 | -5279.1 | 2.41 |
| 36 | NAUS | 47.44 | 5.97 | 574.0 | -3.79 | 722.4 | -5.89 |
| 37 | OHI2 | 5.89 | 2.83 | -1114.5 | -0.53 | -922.6 | -1.03 |
| 38 | ONRJ | 36.33 | 11.44 | -1474.0 | 2.01 | -3058.1 | 3.12 |
| 39 | PALM | 6.78 | 3.03 | -1830.4 | 0.54 | -1815.6 | 0.21 |
| 40 | PARC | 10.24 | 4.04 | -1271.9 | -0.82 | -957.8 | -1.23 |
| 41 | PBCG | 33.55 | 7.50 | 1751.2 | 0.23 | 1541.4 | -0.20 |
| 42 | PEPE | 33.48 | 8.15 | -793.5 | 1.42 | -452.2 | -0.49 |
| 43 | POAL | 26.68 | 11.03 | -548.7 | 0.95 | 430.9 | 0.19 |
| 44 | POLI | 27.42 | 10.39 | 691.8 | -1.11 | 2559.6 | -4.53 |
| 45 | POVE | 50.42 | 8.68 | 100.8 | 0.76 | 95.3 | -0.80 |
| 46 | PPTE | 30.92 | 11.81 | 552.3 | -1.33 | 413.9 | -2.08 |
| 47 | RECF | 38.96 | 7.87 | -684.4 | 2.20 | -167.6 | 2.15 |
| 48 | RIO2 | 9.80 | 3.88 | -814.9 | -0.87 | -274.6 | -1.52 |
| 49 | RIOB | 46.91 | 8.37 | -373.0 | -0.82 | -258.04 | -2.86 |
| 50 | RIOD | 37.82 | 11.60 | -2181.6 | 3.50 | -2835.3 | 3.67 |
| 51 | RNNA | 40.09 | 8.38 | -13.4 | 1.10 | 143.4 | 0.31 |
| 52 | SALU | 47.88 | 6.89 | 312.9 | 0.53 | 154.4 | -1.02 |
| 53 | SANT | 12.51 | 4.91 | -10208.7 | 5.53 | -4790.1 | 4.07 |
| 54 | SAVO | 35.72 | 8.15 | 443.3 | -0.42 | 34.9 | 0.10 |

Table 2: Differences of the mean values of IWV ($\Delta\overline{IWV}$ in [$kg\,m^{-2}$]) between GNSS and the NWM for the period 2007-2013 at 67 stations located in South America and Central America. The mean value ($\overline{IWV}$) from GNSS at each site is also given and SD refers to the standard deviation. $\Delta z = z_{GNSS} - z_{NWM}$ refers to the difference in the geopotential [$m^2 s^{-2}$] at each GNSS station.

| | | **GNSS** | | **ERA-Interim** | | **MERRA-2** | |
| | Name | $\overline{IWV}$ | SD | $\Delta z$ | $\Delta\overline{IWV}$ | $\Delta z$ | $\Delta\overline{IWV}$ |
| | | [$kg\,m^{-2}$] | [$kg\,m^{-2}$] | [$m^2 s^{-2}$] | [$kg\,m^{-2}$] | [$m^2 s^{-2}$] | [$kg\,m^{-2}$] |
| 55 | SCUB | 37.84 | 9.85 | -913.5 | 0.00 | -1402.9 | 0.96 |
| 56 | SMAR | 25.92 | 11.46 | -982.4 | 0.46 | -862.1 | 0.11 |
| 57 | SSA1 | 36.80 | 8.36 | -370.5 | 0.72 | -877.1 | 1.30 |
| 58 | SSIA | 36.46 | 8.42 | 1832.3 | -3.34 | 925.1 | -3.92 |
| 59 | TOPL | 40.69 | 11.33 | -61.1 | -0.05 | -712.3 | -0.22 |
| 60 | TUCU | 25.36 | 12.17 | -4563.5 | 0.51 | -6848.5 | 4.31 |
| 61 | UBER | 28.03 | 10.83 | 638.2 | -2.24 | 600.0 | -3.36 |
| 62 | UCOR | 18.77 | 9.78 | -1912.7 | -0.93 | -2843.9 | 0.32 |
| 63 | UFPR | 23.77 | 9.64 | 2179.4 | -2.78 | 1364.4 | -3.91 |
| 64 | UNRO | 21.74 | 10.52 | 82.5 | -0.64 | 195.0 | -0.51 |
| 65 | UNSA | 19.71 | 10.01 | -7652.7 | 2.50 | -7232.9 | 3.68 |
| 66 | VESL | 3.17 | 1.29 | 2729.8 | 1.03 | 773.3 | 1.07 |
| 67 | VITH | 39.08 | 8.75 | 405.4 | -0.72 | 442.9 | -0.74 |

Table 3: Mean values of the difference between $IWV_{ERAInterim}$ data and $IWV$ computed from the numerical integral of the equation 7 at each grid point surrounding the GNSS site. The integration limits range from 1 hPa to the static geopotential value assigned by ERA Interim to the point ($z^k_{ERAInterim}$).

| Station name | NorthWest | | NorthEast | | SouthWest | | SouthEast | |
|:---:|:---:|:---:|:---:|:---:|:---:|:---:|:---:|:---:|
| | mean | sd | mean | sd | mean | sd | mean | sd |
| ACYA | -0.37 | 0.39 | -0.19 | 0.34 | -0.49 | 0.36 | -0.39 | 0.36 |
| AREQ | 0.13 | 0.74 | 0.15 | 0.53 | -0.12 | 0.32 | 0.00 | 0.33 |
| AUTF | 0.04 | 0.03 | 0.05 | 0.07 | 0.04 | 0.04 | 0.04 | 0.04 |
| AZUL | 0.06 | 0.06 | 0.06 | 0.07 | 0.06 | 0.06 | 0.06 | 0.09 |
| BELE | 0.15 | 0.08 | 0.17 | 0.1 | 0.15 | 0.09 | 0.19 | 0.18 |
| BOAV | 0.15 | 0.12 | 0.15 | 0.08 | 0.15 | 0.14 | 0.14 | 0.08 |
| BOGT | -0.21 | 0.26 | -0.05 | 0.12 | 0.14 | 0.11 | 0.08 | 0.11 |
| BRAZ | 0.14 | 0.09 | 0.14 | 0.09 | 0.15 | 0.11 | 0.14 | 0.09 |
| BRFT | 0.14 | 0.09 | 0.14 | 0.14 | 0.14 | 0.10 | 0.12 | 0.16 |
| BRMU | 0.11 | 0.09 | 0.12 | 0.08 | 0.09 | 0.09 | 0.11 | 0.08 |
| BYSP | 0.14 | 0.08 | 0.15 | 0.08 | 0.13 | 0.08 | 0.13 | 0.09 |
| CEFE | 0.08 | 0.13 | 0.12 | 0.08 | 0.09 | 0.22 | 0.14 | 0.11 |
| CHET | 0.14 | 0.11 | 0.15 | 0.12 | 0.14 | 0.09 | 0.16 | 0.24 |
| CHPI | 0.00 | 0.23 | 0.03 | 0.12 | 0.05 | 0.09 | 0.08 | 0.08 |
| CONZ | 0.05 | 0.05 | 0.02 | 0.07 | 0.03 | 0.05 | 0.00 | 0.10 |
| COPO | -0.02 | 0.07 | -0.92 | 0.53 | -0.03 | 0.07 | -0.55 | 0.52 |
| CRO1 | 0.16 | 0.08 | 0.17 | 0.11 | 0.15 | 0.09 | 0.16 | 0.09 |
| CUCU | 0.21 | 0.20 | 0.55 | 0.22 | 0.01 | 0.29 | 0.56 | 0.40 |
| CUIB | 0.17 | 0.09 | 0.12 | 0.14 | 0.15 | 0.09 | 0.15 | 0.08 |
| EBYP | 0.11 | 0.22 | 0.11 | 0.13 | 0.11 | 0.13 | 0.11 | 0.21 |
| FALK | 0.04 | 0.03 | 0.04 | 0.05 | 0.04 | 0.03 | 0.04 | 0.08 |
| GUAT | 0.04 | 0.24 | -0.17 | 0.24 | 0.06 | 0.14 | 0.00 | 0.11 |
| IGM1 | 0.07 | 0.08 | 0.07 | 0.11 | 0.09 | 0.08 | 0.08 | 0.07 |
| ISPA | 0.09 | 0.08 | 0.09 | 0.09 | 0.10 | 0.08 | 0.09 | 0.12 |
| LPAZ | 0.05 | 0.07 | 0.03 | 0.08 | 0.05 | 0.09 | 0.03 | 0.10 |
| LPGS | 0.08 | 0.08 | 0.04 | 0.10 | 0.13 | 0.23 | 0.09 | 0.16 |
| MABA | 0.18 | 0.10 | 0.17 | 0.09 | 0.16 | 0.09 | 0.17 | 0.21 |
| MANA | 0.16 | 0.07 | 0.17 | 0.07 | 0.15 | 0.23 | 0.13 | 0.16 |
| MAPA | 0.14 | 0.07 | 0.14 | 0.09 | 0.15 | 0.09 | 0.14 | 0.19 |

Table 3: Mean values of the difference between $IWV_{ERAInterim}$ data and $IWV$ computed from the numerical integral of the equation 7 at each grid point surrounding the GNSS site. The integration limits range from 1 hPa to the static geopotential value assigned by ERA Interim to the point ($z^k_{ERAInterim}$).

| Station name | NorthWest | | NorthEast | | SouthWest | | SouthEast | |
|:---:|:---:|:---:|:---:|:---:|:---:|:---:|:---:|:---:|
| | mean | sd | mean | sd | mean | sd | mean | sd |
| MARA | 0.16 | 0.06 | 0.16 | 0.13 | 0.12 | 0.07 | 0.21 | 0.34 |
| MDO1 | 0.06 | 0.06 | 0.11 | 0.22 | 0.05 | 0.06 | 0.04 | 0.07 |
| MERI | 0.17 | 0.24 | 0.11 | 0.18 | 0.19 | 0.11 | 0.14 | 0.09 |
| MGBH | 0.13 | 0.11 | 0.10 | 0.15 | 0.12 | 0.11 | 0.12 | 0.09 |
| MSCG | 0.12 | 0.09 | 0.12 | 0.12 | 0.12 | 0.09 | 0.10 | 0.18 |
| MZAC | -0.53 | 0.63 | 0.05 | 0.16 | -0.19 | 0.53 | 0.04 | 0.16 |
| NAUS | 0.17 | 0.09 | 0.19 | 0.13 | 0.15 | 0.08 | 0.15 | 0.09 |
| OHI2 | 0.00 | 0.04 | 0.01 | 0.02 | 0.02 | 0.01 | 0.02 | 0.01 |
| ONRJ | 0.10 | 0.08 | 0.09 | 0.08 | 0.11 | 0.11 | 0.08 | 0.09 |
| PALM | 0.01 | 0.05 | 0.00 | 0.07 | -0.01 | 0.04 | -0.02 | 0.04 |
| PARC | 0.04 | 0.02 | 0.04 | 0.02 | 0.04 | 0.02 | 0.04 | 0.07 |
| PBCG | 0.15 | 0.12 | 0.13 | 0.11 | 0.16 | 0.14 | 0.13 | 0.14 |
| PEPE | 0.16 | 0.10 | 0.19 | 0.18 | 0.17 | 0.11 | 0.16 | 0.10 |
| POAL | 0.12 | 0.13 | 0.08 | 0.07 | 0.17 | 0.27 | 0.07 | 0.08 |
| POLI | 0.12 | 0.08 | 0.11 | 0.08 | 0.08 | 0.16 | 0.00 | 0.18 |
| POVE | 0.17 | 0.14 | 0.19 | 0.20 | 0.17 | 0.14 | 0.17 | 0.13 |
| PPTE | 0.12 | 0.10 | 0.12 | 0.08 | 0.11 | 0.12 | 0.11 | 0.08 |
| RECF | 0.09 | 0.11 | 0.09 | 0.11 | 0.12 | 0.17 | 0.12 | 0.13 |
| RIO2 | 0.03 | 0.03 | 0.04 | 0.02 | 0.02 | 0.06 | 0.02 | 0.02 |
| RIOB | 0.14 | 0.11 | 0.14 | 0.18 | 0.15 | 0.15 | 0.13 | 0.09 |
| RIOD | 0.10 | 0.08 | 0.10 | 0.08 | 0.12 | 0.13 | 0.11 | 0.10 |
| RNNA | 0.09 | 0.10 | 0.10 | 0.12 | 0.13 | 0.09 | 0.13 | 0.09 |
| SALU | 0.16 | 0.09 | 0.15 | 0.10 | 0.17 | 0.10 | 0.14 | 0.17 |
| SANT | -0.30 | 0.48 | -0.03 | 0.23 | -0.42 | 0.45 | 0.00 | 0.23 |
| SAVO | 0.13 | 0.19 | 0.16 | 0.17 | 0.13 | 0.10 | 0.15 | 0.09 |
| SCUB | 0.12 | 0.23 | 0.13 | 0.11 | 0.14 | 0.07 | 0.14 | 0.07 |
| SMAR | 0.12 | 0.16 | 0.12 | 0.18 | 0.10 | 0.08 | 0.10 | 0.09 |
| SSA1 | 0.16 | 0.12 | 0.14 | 0.19 | 0.12 | 0.10 | 0.15 | 0.09 |
| SSIA | 0.06 | 0.12 | 0.08 | 0.11 | -0.01 | 0.15 | 0.00 | 0.25 |

Table 3: Mean values of the difference between $IWV_{ERAInterim}$ data and $IWV$ computed from the numerical integral of the equation 7 at each grid point surrounding the GNSS site. The integration limits range from 1 hPa to the static geopotential value assigned by ERA Interim to the point ($z^k_{ERAInterim}$).

| Station name | NorthWest | | NorthEast | | SouthWest | | SouthEast | |
|:---:|:---:|:---:|:---:|:---:|:---:|:---:|:---:|:---:|
| | mean | sd | mean | sd | mean | sd | mean | sd |
| TOPL | 0.18 | 0.09 | 0.19 | 0.16 | 0.18 | 0.10 | 0.17 | 0.12 |
| TUCU | 0.11 | 0.11 | 0.14 | 0.43 | 0.12 | 0.09 | 0.19 | 0.21 |
| UBER | 0.15 | 0.14 | 0.15 | 0.08 | 0.16 | 0.15 | 0.15 | 0.08 |
| UCOR | 0.09 | 0.10 | 0.07 | 0.09 | 0.07 | 0.16 | 0.04 | 0.22 |
| UFPR | 0.02 | 0.17 | 0.04 | 0.10 | 0.03 | 0.11 | 0.03 | 0.09 |
| UNRO | 0.08 | 0.08 | 0.06 | 0.08 | 0.08 | 0.08 | 0.05 | 0.07 |
| UNSA | -0.27 | 0.25 | -0.63 | 0.43 | -0.55 | 0.32 | -0.59 | 0.52 |
| VESL | -0.03 | 0.05 | -0.03 | 0.05 | -0.02 | 0.04 | -0.02 | 0.05 |
| VITH | 0.15 | 0.10 | 0.17 | 0.09 | 0.13 | 0.15 | 0.15 | 0.09 |

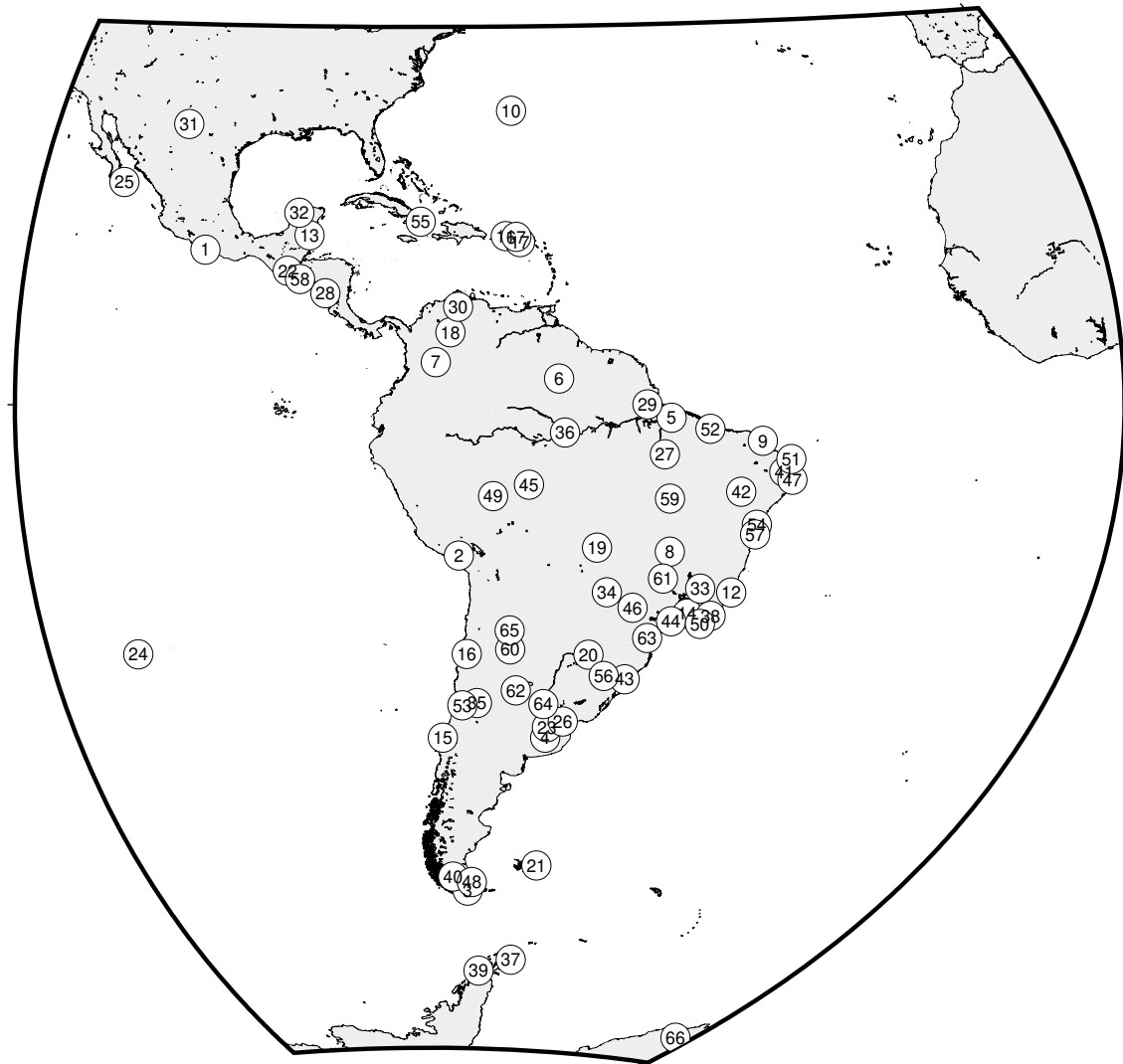

**Figure 1.** Location of the GNSS stations (see Tables 1 and 2).

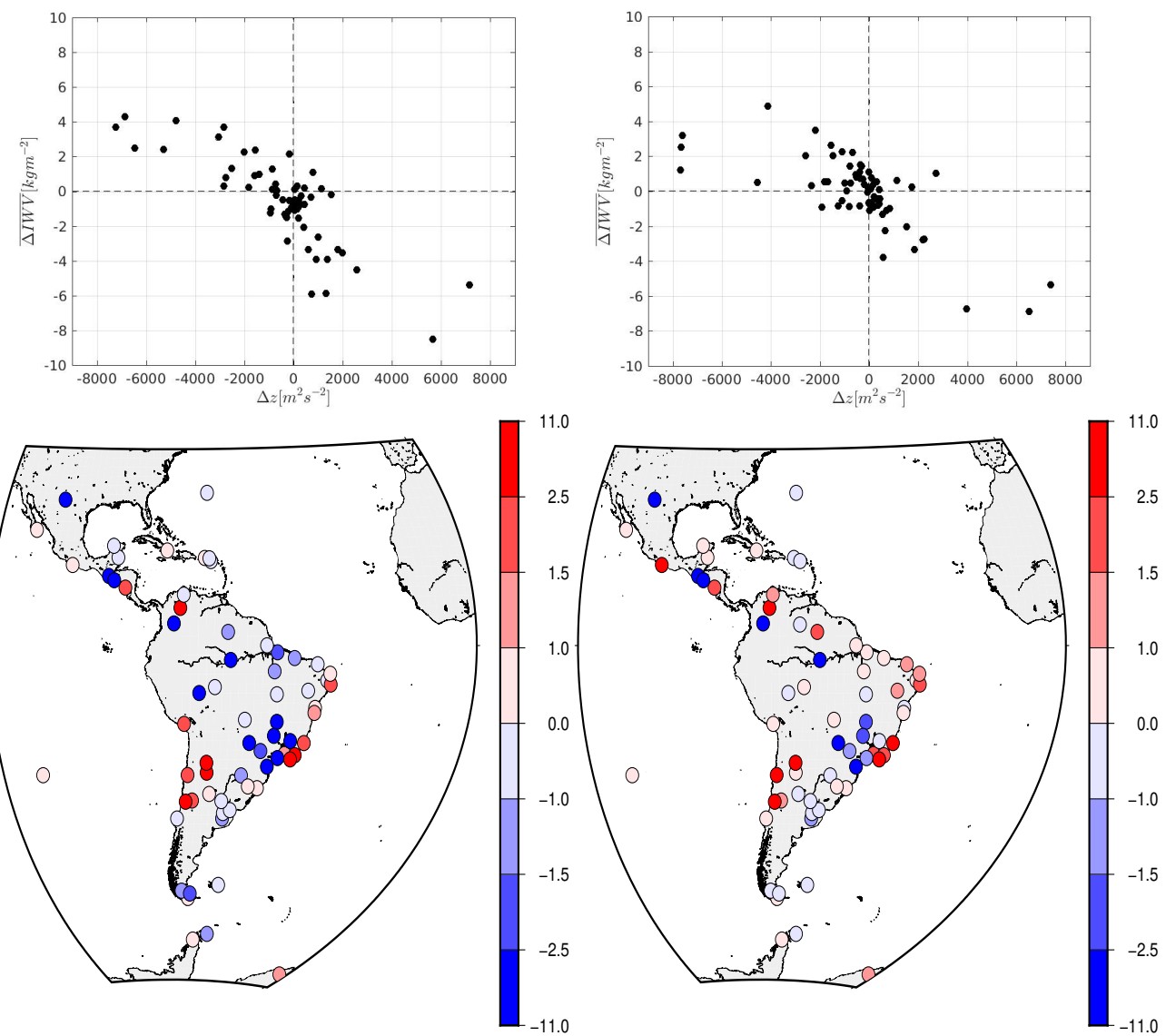

**Figure 2.** (up) Values of $\Delta\overline{IWV}$ ($\Delta\overline{IWV} = \overline{IWV}_{GNSS} - \overline{IWV}_{NWM}$) as a function of the geopotential differences $\Delta z$. Results for MERRA-2 are on the left and the same for ERA Interim on the right. (down) Geographical distribution of $\Delta\overline{IWV}$ for MERRA-2 (left) and ERA Interim (right).

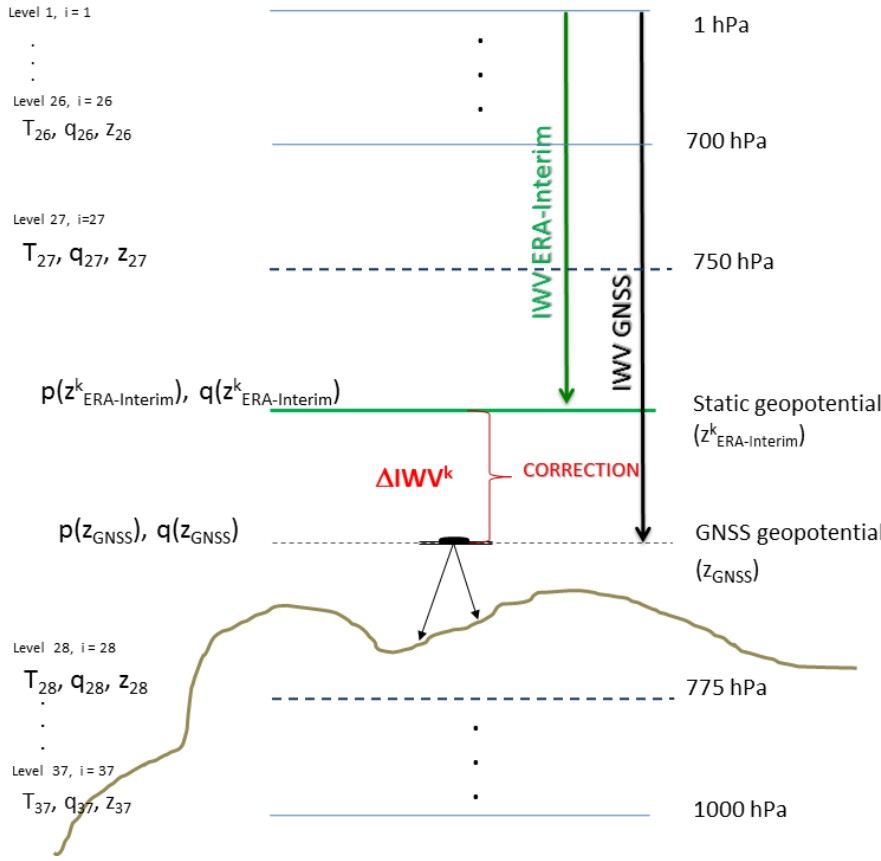

**Figure 3.** Scheme of the applied correction to the IWV from ERA-Interim reanalysis for one of the four grid points (k) at a given instant. Both unknowns ($z_{GNSS}$, dark grey dashed line and $z_{NWM}^k$, thick green line) are located between the pressure levels 27 (750 hPa) and 28 (775 hPa) indicated with thick dashed lines. Atmospheric pressure ($p_i$), temperature ($T_i$), specific humidity ($q_i$) and geopotential ($z_i$) are known in the 37 levels from 1 hPa till 1000 hPa. Values of $p(z_{ERA-Interim}^k)$ and $p(z_{GNSS})$ are calculated from equation (5), whereas the values of $q(z_{ERA-Interim}^k)$ and $q(z_{GNSS})$ resulted from a linear interpolation. The computation of $\Delta IWV^k$ must be done four times (k = 1,...,4), prior to the bi-linear interpolation that produces $\Delta IWV_{ERA-Interim}$ for the given instant.

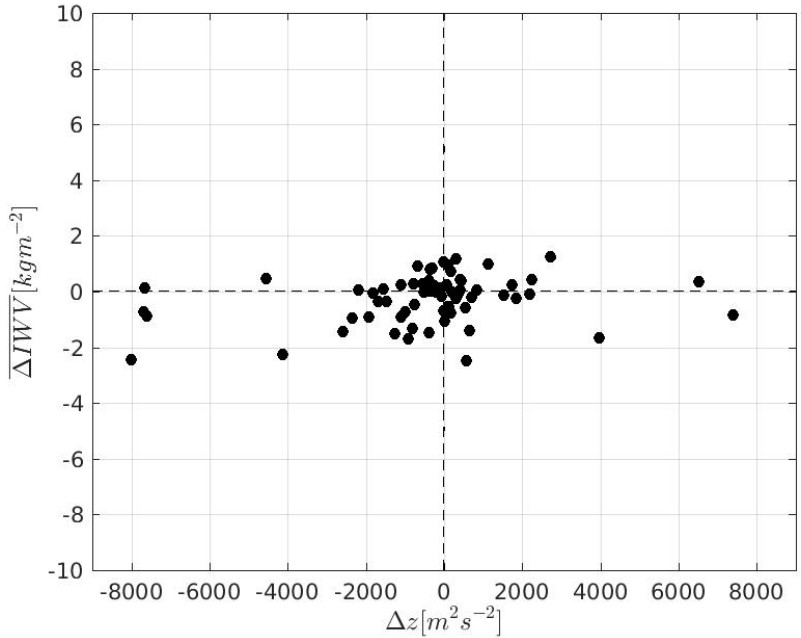

**Figure 4.** Values of $\Delta\overline{IWV}$ ($\Delta\overline{IWV} = \overline{IWV}_{GNSS} - \overline{IWV}_{ERAInterim}$) as a function of $\Delta z$ after applying the proposed correction.

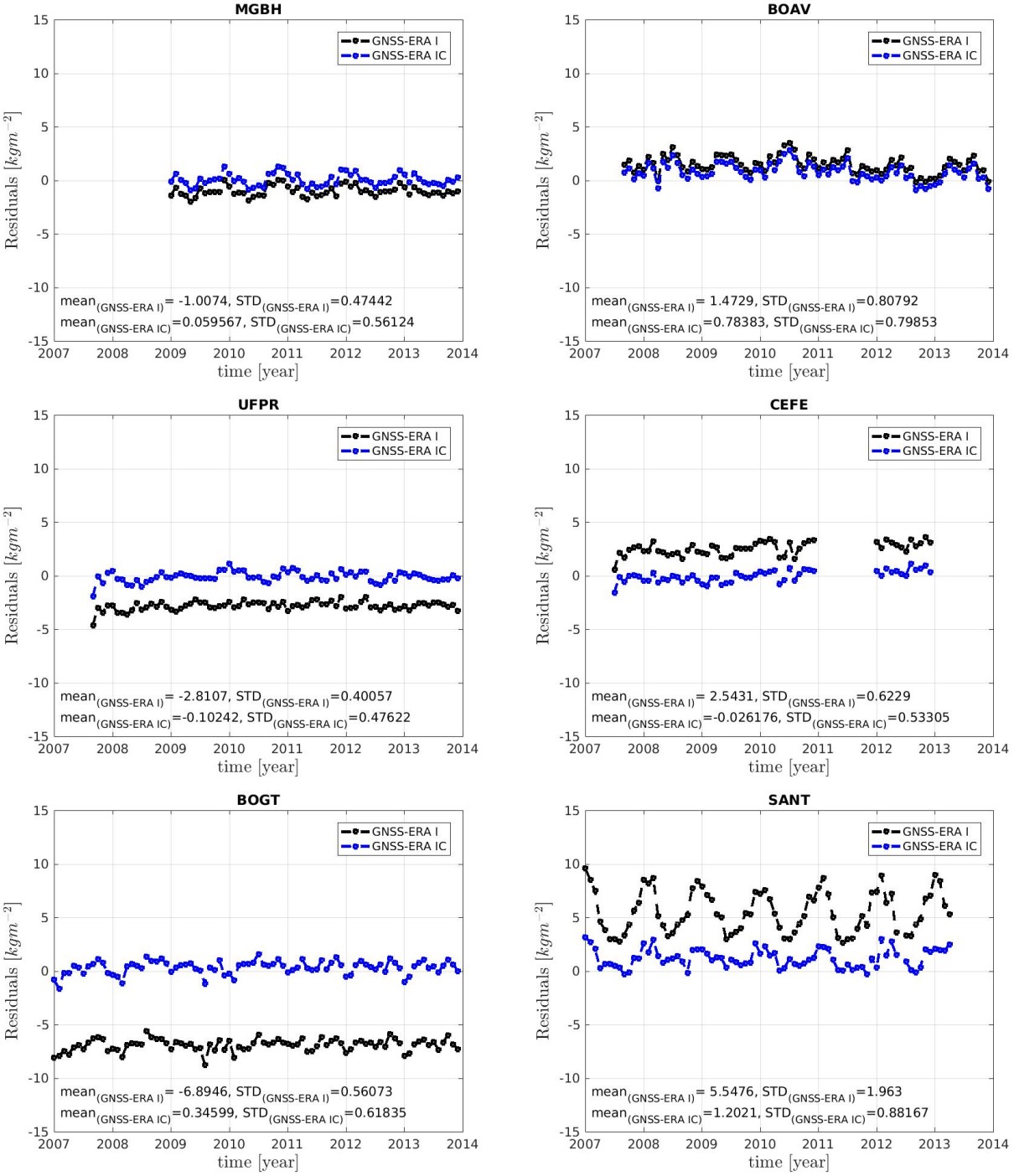

**Figure 5.** Residuals of the difference $(IWV_{GNSS} - IWV_{ERAInterim})$ (GNSS - ERA I, black line) along with residuals of the difference $[IWV_{GNSS} - (IWV_{ERAInterim} + correction)]$ (GNSS - ERA IC, blue line). Left column shows stations with positive $\Delta z$, it means that GNSS station is higher to the location assigned by ERA-Interim, and the opposite is at the right column. Mean values of the residuals along with the standard deviations are also provided. The sites are shown according to $\Delta z$ decreasing from top to bottom at each column.

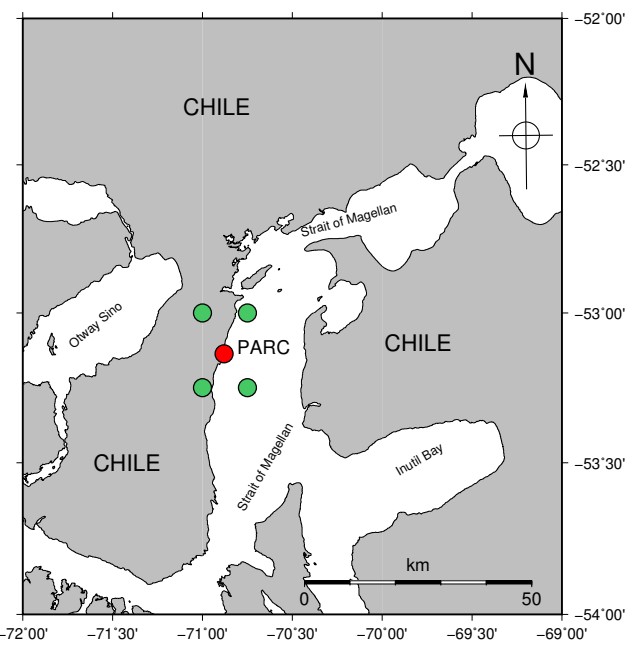

**Figure 6.** Location of GNSS station PARC along with the 4 grid points around the station. The grid points correspond to ERA-Interim.