# Peer review of "A numerical method to improve the spatial interpolation of water vapor from numerical weather models: a case study in South and Central America"

_Annales Geophysicae, 2019_

## Referee Comment (RC1) · Anonymous Referee #1 · 14 Mar 2019

**General Comments:**

In this paper integrated water vapour (IWV) data from GNSS are compared to data from two numerical weather models (ERA-Interim from ECMWF and MERRA-2 from NASA). In this context, a correction for differences in surface geopotential heights is proposed and successfully applied. Except for some details mentioned below the methods used and the results of the comparisons are well described. My major concerns are related to the height correction as shown in the comments below.

1. Novelty of the correction approach and related results:

[Figure]

I think the application of a surface altitude correction is not new. It it commonly used e.g. in the satellite community if model data are taken as input for a retrieval. However, I must admit that I did not find a proper reference for this – it is (silently) assumed as obvious that such kind of correction is required. Furthermore, it is also obvious that structures smaller than the model spatial solution cannot be resolved and that therefore any correction will be less good in case of sea/land edges, small islands or large topographic changes within one model grid box.

2. Application of correction:
   It should be explained why the correction is not applied to data with small height differences. In this case the correction should be small but still give an improvement, and it would not be necessary to have a (somewhat artificial) criterium on $\Delta Z$ to decide if the correction should be applied or not.

3. Definition of correction:
   At several places in the manuscript it is mentioned that the correction for different surface geopotential heights is subtracted or added to the data depending on the sign of the height difference. It would be much clearer to have corrections with different signs and a fixed definition on how the correction is applied (either added or subtracted to $IWV_{GNSS}$ or $IWV_{NWM}$). This would also be more consistent with the results (positive and negative differences) shown e.g. in Fig. 4.

4. Computation of correction (section 3.2):
   As explained in section 3.2 the correction is derived from an integration of (interpolated) NWM profiles over the geopotential height difference range. Especially for smaller height differences this integration should depend on the vertical interpolation (or maybe even extrapolation) of the profiles. A good indication for a valid integration would be a comparison between the gridded IWV and the corresponding integral over $q$ at a model grid point. Since for the correction $q$ and $T$ profiles are interpolated to the GNSS station position, why not determine the

IWV directly from the integral of $q$ over height (starting at station height)? This would not require additional IWV grid information from the NWMs. This should be discussed.

5. Temporal interpolation:
It seems that model data are only spatially interpolated to the locations of the GNSS stations. How are temporal differences handled? Are only GNSS data used close to the model grid times (assuming which maximum time difference)? How do temporal differences affect the results? Please explain.

**Specific Comments:**

1. p. 1, l. 22–23:
"the correction procedure is not advisable either for a coastal station and/or stations in islands"

Actually, islands are not specifically mentioned in the main part of the manuscript. This should be aligned.

2. p. 2, l. 22:
"That is because of a misrepresentation of ECMWF analysis"

This formulation is unclear. This sounds like the authors did something wrong in interpreting the ECMWF data which is probably not what is meant here. Please clarify.

3. p. 3, l. 21–22:
"resulted from a geodetic process of (GPS + GLONASS) data"

What is meant with "geodetic process" - please explain. Alternatively, you could remove this part of the sentence if you just refer to the data product from Bianchi et al. (2016a).

4. p. 4, section 2.1: Also here the "geodetic process" is mentioned several times without explanation. Other formulations are unclear and terms remain unexplained, e.g. (l. 13): "The GNSS observations were processing at a double-difference level" or (l. 17–18): "The comparison of ZTD results shows the expected consistency between estimations from the homogeneous but independent analysis." What does the latter mean for the quality of the data product?

It seems that the authors use a previously published GNSS data product for the study. Therefore I suggest to reformulate this section such that it mainly refers to this underlying study and only explains the principle of the retrieval method and the basic properties of the data set (like temporal resolution, expected accuracy) without going into details.

5. p. 6, eq. 5:
What is the pressure calculated by this formula used for? Is it for the integration of $q$ in eq. 6? The NWMs should also provide pressure (consistenly with $q$) - why are these data not used here? Is the lapse rate $\lambda$ assumed to be independent from height? If yes, for which altitude range is this assumption valid?

6. p. 7, l. 22:
What is meant with "inter-annual averages"? Do you mean an average over the complete time series? Please clarify.

7. Section 4.1:
I suggest to add plots of the content from Table 2 (e.g. station vs. IWV$\pm$SD for the different data sets, possibly $\Delta Z$ vs. IWV) to facilitate the interpretation of the results.

8. p. 8, l. 8:
"model failures" is the wrong term here. You cannot expect from a model that it provides perfect results for points not on the model grid. Please reformulate.

[Figure]

9. p. 8, l. 9:
   "...the disagreement is the greatest for the stations classified as Critical"

   This statement is obvious, the sentence can be deleted.

10. p. 9, l. 29 (and eq. 5): Is $\delta Z$ and $\Delta Z$ the same quantity? If not, explain the difference, otherwise adapt notation.

11. p. 10, l. 3:
    "MERRA-2 generally overestimates $IWV_{GNSS}$"

    Is this a new finding (in this case it should be emphasised more) or something already known (then give references)?

12. p. 18, Fig. 1:
    I think this figure is not absolutely necessary as it just visualises the bi-linear interpolation which is a standard method.

**Technical Corrections:**

1. p. 1, l. 22:
   grid point → grid points

2. p. 2, l. 3:
   Keyboards → Keywords

3. p. 4, l. 9:
   come → comes

4. Section 2.1.1 should probably be section 2.2

5. p. 8, l. 1:
   fulfill → fulfills

6. p. 8, l. 20:
   exceed → exceeds

7. p. 8, l. 23:
   better than a → better than

8. p. 8, l. 24:
   have to be added → has to be added

9. p. 21, Fig. 4:
   The y axis of the plots should be labelled (e.g. "IWV difference") and not only
   show a unit.
* * *

---

## Referee Comment (RC2) · Anonymous Referee #2 · 1 Apr 2019

[referee-annotated manuscript omitted]

---

## Author Comment (AC1) · 14 Jun 2019

The authors would like to thank both anonymous reviewers for their contributions, which have enriched our work. We have taken all their comments and suggested corrections and we have completely changed the manuscript in the title and structure as well as in the organization and quantity of contents and results we had shown.

In brief we enumerate the most important modifications present in this new version of the manuscript:

a) the classification of the stations following the geopotential height difference (small,

␟

large and critical) was dismissed and the complete set of stations was analyzed as a whole. Thus, new tables, figures and plots were adequate to this.

b) Geopotential heights were changed by geopotentials [m2 s-2] and the nomenclature was also changed: z lower case instead of z upper case.

c) Figure 1 was eliminated

d) New table 1 shows geopotential GNSS and the static geopotential values assigned by the models to each GNSS site. The geopotential for ERA Interim and geopotential for MERRA2 come from a bi-linear interpolation of the given static geopotential values at the 4 grid points surrounded the GNSS site.

e) A discussion about the behavior of the mean IWV from the reanalysis models with respect to the mean IWVGNSS highlights overestimations and underestimations is incorporated. New plots are also incorporated to easily follow the discussion of the new findings.

f) A new Table 3 was included in order to demonstrate the robustness of our numerical integration method for reproducing IWV values at ERA Interim grid points around each GNSS site. For this calculation we used the q and t data (specific humidity and temperature) given at 37 atmospheric pressure levels. This q, t and p set is the same data used for the calculation of the integral correction.

g) Likewise, and following the suggestion, new figures were incorporated to improve the visualization of the results of the comparison between the models and GNSS, prior to the application of the integral correction.

h) The scheme of application of the correction for a given example was clarified in its caption and through new text incorporated in the main body of the manuscript.

i) The correction is presented with a new equation independently of the integral definition of the IWV. Moreover, the different possible signs for the correction are included in this new mathematical expression.

j) The previous classification by height differences (small, large, critical) is sketched out without mentioning it in the new presentation of the results. The residuals of the differences (IWV GNSS -IWV ERA Interim) before and after applying the integral correction are shown in a new figure. The new figure also shows the results for cases where the model geopotential is located above the GNSS geopotential (right column) and below the GNSS potential (left column).

k) Also following the suggestion, the title was changed since the region of South and Central America only refers to the GNSS sites available for this work and we do not perform any analysis of the IWV behavior in the region.

Following, the detailed answers to the reviewer: Answers to Anonymous Referee # 1:

Application of the correction This comment was considered and the integral correction strategy was applied to the whole set of data. Effectively, as you affirmed, the correction applied to the stations formerly classified as "small" is slight but still it is an improvement.

Definition of the correction The correction was defined independently of the integral definition of IWV. Both negative and positive results are included in equation (7) because the sign is given by the difference between atmospheric pressure values (PGNSS - PNWM). For a sake of clarity some paragraph were also included and a better explanation of the example (now Figure 3) is also given.

Computation of the correction According to the recommendations received by both reviewers, the structure and presentation of the work has changed. We have placed in the methodology section: the calculation of the GNSS geopotential from the geodetic coordinates of the station, the comparison of the mean values of both models with respect to the mean values IWVGNSS, as well as the quantification of the geopotential differences and a brief summary of the method for calculating the correction. The details of the calculation of the correction are presented in the following section and finally the results section only presents results after having applied the correction. Thus, the

way we compute and applied the proposed correction was clarified in the main text. Moreover, the suggestion of this reviewer was taken into consideration and the numerical integration procedure was tested for the whole set of stations. In the new Table 3 the mean values of the difference IWV from ERA Interim and the same IWV from a numerical integration of over q at each grid point is shown. The integral is computed from 1 hPa till the static geopotential height at each grid point and we used data given at 37 pressure levels from ERA Interim. Each of the 4 columns correspond to the 4 grid-point around the GNSS station. The averages and standard deviations were computed over the period 2007-2013. In addition, we have also calculated the alternative suggested by this reviewer: We have computed the integral over q from 1 hPa till the geopotential corresponding to GNSS at the 4 grid points surrounding the GNSS station. Then the value at the GNSS site was calculated using a bi-linear interpolation. However, given that the results proved to be very similar to our procedure (both the mean values and their dispersions), we have decided to omit them in favour of the extension of the work and given that this strategy does not add up different results. Note that this strategy differs from the integral performed at grid points from 1 hPa to the static geopotential of each point. These results were incorporated as before mentioned in Table 3.

Temporal interpolation: A paragraph was included to explain how the different time intervals of the datasets were handled.

Specific comments:

1. L. 22-23 abstract The discussion was included in the main part of the manuscript

2. P. 2 L. 22 Corrected. A new sentence was added

3. P. 3 L 21-22 and P. 4 section 2.1 Following your advise we just explain the main characteristics of the data set and removed the incomplete presentation, we also refer the reader to the work from Bianchi et al, 2016a for further technical details.

5. P. 6 eq. 5 The application of equation 5 is clarified in the text. This is the necessary

formula to estimate the atmospheric pressure p at zGNSS as well as at the geopotential of the each grid point around the GNSS site. These geopotentials (GNSS and the 4 grid points) are not necessarily coincident (generally they are not) with the geopotential correspondent to the 37 given pressure levels. As a matter of fact temperature (T) and pressure (p) data at each level are necessary to compute the p unknown at each geopotencial by using eq. 5. The unknown temperature at these geopotentials is estimated by assuming the rate 0.006499 °K/m. Thus, the unknown temperature is given by the numerator of Eq. 5.

6. P. 7 L 22 Yes, "interannual" averages refer to the mean value over the complete period 2007-2013. The sentence was clarified and this terminology avoided.

7. Section 4.1 Following your suggestion the tables were reworked and also graphics were added to enrich the comparison. Thank you.

8. P. 8 L. 8 The expression "model failure" was eliminated. The section was rewritten.

9. P. 8, L 9 This part was removed. The classification in: small, critical and large was dismissed.

10. P 9 L. 29 (and eq. 5) The methodology section was rewritten and it includes the explanation of $\Delta z$. On the other hand the meaning of $\delta z$, within equation 5, was clarified.

11. P. 10. L. 3 We emphasise this point with more discussion and a new figure

12. P 18 The figure was removed

Technical corrections

1. P1 L. 22 The abstract was rewritten.

2. P. 2 L. 3 Corrected

3. P.4 L.9 removed from the main text

4. Section 2.1.1 should probably be section 2.2 Corrected

5. to 8. These parts were eliminated from the main text

9. P. 21 former Fig 4 This figure was eliminated since its purpose was to show the behavior of the stations classified as small for not applying there the correction.

Please also note the supplement to this comment:
https://www.ann-geophys-discuss.net/angeo-2019-20/angeo-2019-20-AC1-supplement.zip

---

## Author Comment (AC2) · 14 Jun 2019

The authors would like to thank both anonymous reviewers for their contributions, which have enriched our work. We have taken all their comments and suggested corrections and we have completely changed the manuscript in the title and structure as well as in the organization and quantity of contents and results we had shown.

In brief we enumerate the most important modifications present in this new version of the manuscript:

a) the classification of the stations following the geopotential height difference (small,

large and critical) was dismissed and the complete set of stations was analyzed as a whole. Thus, new tables, figures and plots were adequate to this.

b) Geopotential heights were changed by geopotentials [m2 s-2] and the nomenclature was also changed: z lower case instead of z upper case.

c) Figure 1 was eliminated

d) New table 1 shows geopotential GNSS and the static geopotential values assigned by the models to each GNSS site. The geopotential for ERA Interim and geopotential for MERRA2 come from a bi-linear interpolation of the given static geopotential values at the 4 grid points surrounded the GNSS site.

e) A discussion about the behavior of the mean IWV from the reanalysis models with respect to the mean IWVGNSS highlights overestimations and underestimations is incorporated. New plots are also incorporated to easily follow the discussion of the new findings.

f) A new Table 3 was included in order to demonstrate the robustness of our numerical integration method for reproducing IWV values at ERA Interim grid points around each GNSS site. For this calculation we used the q and t data (specific humidity and temperature) given at 37 atmospheric pressure levels. This q, t and p set is the same data used for the calculation of the integral correction.

g) Likewise, and following the suggestion, new figures were incorporated to improve the visualization of the results of the comparison between the models and GNSS, prior to the application of the integral correction.

h) The scheme of application of the correction for a given example was clarified in its caption and through new text incorporated in the main body of the manuscript.

i) The correction is presented with a new equation independently of the integral definition of the IWV. Moreover, the different possible signs for the correction are included in this new mathematical expression.

j) The previous classification by height differences (small, large, critical) is sketched out without mentioning it in the new presentation of the results. The residuals of the differences (IWV GNSS -IWV ERA Interim) before and after applying the integral correction are shown in a new figure. The new figure also shows the results for cases where the model geopotential is located above the GNSS geopotential (right column) and below the GNSS potential (left column).

k) Also following the suggestion, the title was changed since the region of South and Central America only refers to the GNSS sites available for this work and we do not perform any analysis of the IWV behavior in the region.

Following, the detailed answers to each of the reviewer: Anonymous Referee # 2

Reviewer #2 made all comments and corrections in the text. Because the main text has changed dramatically, we will answer here the questions that need further explanation since the grammatical errors disappeared when rewriting or eliminate those parts of the text.

Page 3: #5: vague statement. The exact quantity of years was included in the text

Page 3: #6: in Geodesy, we usually designated H for geopotential height and Z for the third component of the Cartesian coordinate system Yes, it is true but some authors also designate H for the orthometric height in order to distinguish it from h the ellipsoidal height. Therefore, we decided to adopt z (lower case) and express the differences in terms of geopotential (not geopotential height). In this way, we use the data from the models as they are provided (geopotential in m2/s2) and only the GNSS height has to be converted.

Page 3: #8-9 why 100 m and not 90 m, 110 or another value? These comments were taken into account and the entire available dataset was studied without discrimination.

Page 4: #1 to #4. The description of the geodetic processing was incomplete and resulted unclear. Because we used IWV from GNSS from a previously published study,

we reformulate the section including just the reference of the source and the mean characteristics of the dataset.

Page 4: #5 A mention to the partial evaluation of MERRA2 was included.

Page 5: #1.to #11 ; Page 6: #5 to #7; Page 7: #4 to #6, #8 The sections Methodology and the subsection Computation of the integral correction were rewritten. For a sake of clarity, the different paragraphs were reordered and some other sentences added. In this new text we took into account the items highlighted by the reviewer: A clarification of how the geopotential GNSS was calculated from geodetic data, An explanation about how the geopotential GNSS (zGNSS) and the static geopotential data from the models at the 4 grid points (ziNWM) are related. We also explained how we computed p, t and q at zGNSS and at ziNWM . Or in other words, an explanation of how the formulas were used. We also described how the correction is calculated and how to take into account the sign of the correction. We also highlighted which is the difference between $\Delta z$ and $\delta z$. Finally, A more detailed description of the example (see Figure 2) was included

Page 6: #1 and #2 The former discrimination in small, large and critical height differences was dismissed in this new manuscript.

Page 6: #3 and #4 Given that any structure smaller than the resolution of the model could not be evidenced and considering that many of the GNSS stations of the available dataset are in mountain areas, the model with the smallest grid was chosen. It is expected that stations located near or at mountainous regions will suffer great height changes in short distances. We assume that the model with the finest grid can better reflect this situation. Moreover, we better explained why we also took into account results from Zhu et al. (2014) to back up this decision.

Page 7: #7 The suggested reference was incorporated

Page 7: #9 to #11; Page 8: #7, Page 9: #1, Page 10: #1 The section Results was

rewritten and now it incorporates the old section Application of the integral correction. Then, it includes only the results after the application of the integral correction. On the other hand, the comparison between IWVGNSS and IWVNWM was moved to the section Methodology. The title was changed.

Page 10: #2 The section discussion and conclusions was rewritten too. The agreement with the state-of-the-art literature was also highlighted.

About originality of the work: although the application of an altitude correction is not new, in fact it is commonly accepted and silently assumed, it is not widely studied. In other words, the statistical quantification of the differences between IWV from NWM and GNSS is not extensively known. In this paper we offer an analysis of the differences that users of IWV data from NWM in South and Central America might encounter if they intend to use such data as a substitute for IWVGNSS values.

Please also note the supplement to this comment:
https://www.ann-geophys-discuss.net/angeo-2019-20/angeo-2019-20-AC2-supplement.zip

---

## Referee Report (RR1)

**General Comments:**

The revised version of the manuscript has been largely rewritten based on the reviewers' comments. It seems that all of my comments to the previous version have been essentially considered. The paper has improved, but there are several issues in the new version (see below) which need to be addressed by the authors.

My main comment is that the aim of the paper should be pointed out more clearly in the text (and in the title). Especially, it should be clarified that the described procedure is not a correction for NWM data, it is a method to achieve improved results in case of a spatial interpolation of NMW data. I also miss a more quantitative assessment of the results by putting them into relation with previous work and by clarifying what is new/different in the current work.

**Specific Comments:**

1. Title:
   The title has been changed as response to a comment of the other reviewer to:

   "Analysis of the geopotential from NWM at GNSS sites and its influence in IWV computation"

   I think this new title is not clearly formulated. There is no geopotential provided by NWMs at the GNSS sites, this is calculated by the authors. As I understand, the paper also does not analyse geopotential data, they are used in the context of a correction procedure, which shall be applied when interpolating NMW data (specifically IWV) to geolocations not on the NMW spatial grid. The improvement is then shown by comparison with GNSS data. Furthermore, I recommend to avoid acronyms in the title as far as possible.

2. Section 2.2:
   The geopotential data from the NWMs which are used in this study ($z^i_{NWM}$, $z_{level}$) should also be mentioned in this section.

3. p. 4, l. 22–27:
   I think these two paragraphs can be omitted or need to be reformulated because:

   - If you want to mention the difference between geopotential and geopotential height, you should also explain why you use geopotential and not geopotential height (or other height coordinates) in this study.

   - It is rather trivial that the vertical integral of water vapor depends on the vertical range you integrate over.

   - The sentence on selection criteria seems not relevant as the correction is applied to all data.

4. p. 6, under-/overestimation of IWV by NWMs:
   In the interpretation of differences between NMWs and GNSS data the expected accuracy (or variability / SD) of the GNSS data should be mentioned and taken into account (see also suggested modifications of Figs. 2 / 4 below). Are the observed under-/overestimations of IWV significant in this context?

5. p. 7, l. 5–6:
   "Finally, the correlation coefficients between $\overline{IWV}_{GNSS}$ values and the respective ones for both NWM, are higher than 0.95 in most of the cases (not shown)."

   What exactly is meant with "in most of the cases"? When talking about averages, there should be only two cases: GNSS vs. ERA-Interim and GNSS vs. MERRA-2. Or, are you referring to correlations of time series at each station?

6. p. 7, l. 8:
   "Following we proceed to compensate the $IWV_{NWM}$ at each of the grid points."

   This sentence is unclear and needs to be reformulated. The method uses adjusted data at the grid points to provide a better model estimate at the locations of the GNSS stations, but the original grid values are still valid.

7. p. 7, l. 18–20:
   "the value of the geopotential (...) may differ several hundred meters."

   Be consistent with quantities/units. "meters" refers to geopotential height, not geopotential.

8. p. 8, eqs. (7), (8) and related text:
   The formula in Eq. (5) (though correct) seems quite complicated. Why not integrate always from $p_{GNSS}$ to $p_{NWM}$? Then, the sign of the correction would automatically be correct without any $(-1)^n$ factors. This would also make some of the following (somewhat confusing) text about highest/lowest values and positive/negative corrections obsolete.

   I am also not sure why the additional eq. (8) is needed as we are just talking about a numerical (linear) integration of eq. (6) and (as described at the top of page 8) you already have all data interpolated to all required $z$ levels (not only the 37 pressure levels).

   If you want to include eq. (8) as a special case, you need to explain the reasons for this.

9. p. 10, l. 11–18:
   This section refers to related studies, but there is no quantitative comparison of results given. Please specify how your results differ from or agree with the mentioned studies and especially what is the new aspect.

10. Fig. 2:

    Since there is no real dependence between $\Delta z$ and mean IWV, I suggest to modify the two top plots such that they show $\Delta$IWV (possibly with SD error bars) as function of $\Delta z$. Then, over- or underestimations of the interpolated model data could be seen more clearly (see also corresponding comment above).

    Please separate the color bar (for $\Delta$IWV) from the x axis label of the top graphs (IWV) and provide a separate label for the color bar. Please add a grid (or at least a zero line) in the upper plots.

**Technical Corrections:**

1. p. 2, l. 33:
   "sign of IWV bias" $\rightarrow$ "sign of the IWV bias"

2. p. 3, l. 10–11:
   "Follows the explanation of the methodology and the presentation of the results obtained after applying the proposed correction to IWV values from ERA-Interim."

   This is no sentence, please reformulate.

3. p. 4, l. 6–7:
   "MERRA-2 represents a quality improvement compared with MERRA because of the trends and jumps linked to changes in the observing systems."

   This sentence is misleading and should be reformulated. You probably mean that MERRA-2 is improved because it contains less trends and jumps than MERRA.

4. p. 4, l. 11–12:
   "To this application we used two different data sets. First, the gridded values of the vertical Integral of Water Vapor (IWV) from both re-analysis models."

This should be reformulated. The second part is again no sentence, and you use more than two data sets (profiles and maps of different parameters from different NWMs). Probably you want to distinguish here between 2D and 3D (profile) data.

5. p. 4, l. 12–13:
   "Because the comparison is performed at each GNSS station, a bi-linear interpolation of each gridded data set was performed."

   This sentence refers to the methodology (described later) and can be removed.

6. p. 4, l. 14:
   "vertical values" → "vertical profiles"

7. p. 4, l. 15:
   "These variable," → "These variables,"

8. p. 4, l. 28:
   "we followed van Dam et al. (2010) algorithm." → "we followed the van Dam et al. (2010) algorithm."

9. p. 5, l. 3:
   "with a = 6378137m. and b = 6356752.3142m. are the" → "with a = 6378137 m and b = 6356752.3142 m being the"

10. p. 5, l. 7:
    "with $e^2$" → "where $e^2$"

11. p. 5, l. 16–17:
    "corrects its values of $IWV_{NWM}$" → "corrects the values of $IWV_{NWM}$"

12. p. 6, l. 1:
    "On the other hand, The" → "On the other hand, the"

13. p. 6, l. 35:
    "Being these values the result of an average over all the $\Delta IWV$ differences."

    This is again no sentence, please reformulate.

14. p. 7, l. 11:
    "Zhu (2014) compare" → "Zhu (2014) compared"

15. p. 7, l. 14:
    "specific humidity the key variable" → "specific humidity, the key variable"

16. p. 8, l. 5:
    "Because each value" → "Each value"

17. p. 8, l. 16:
    The formulation "If $z_{GNSS}$ and $z_{NWM}^i$ are at the same level" is not correct. You probably mean that $z_{GNSS}$ and $z_{NWM}^i$ are in the same layer (i.e. the range between two adjacent levels).

18. p. 8, l. 27:
    $t \to T$

19. p. 9, l. 7:
    "It can be seen that in general the values are resulted very close to zero." → "It can be seen that in general the resulting values are very close to zero."

20. Table 1 (caption):
    The quantities $z_{NWM}$ and $z_{NWM}^i$ are not defined at this stage (and are not used in the table), so please reformulate.

21. Fig. 3:
    Actually, the line for $z_{GNSS}$ does not appear as a dark blue dashed line, it is more grey.

22. Fig. 4:
    Should be aligned to top figures of Fig. 2 (see comments above). Actually, the left sub-figure in Fig. 4 is the same as the top left sub-figure of Fig. 2, only with a different color scale, so depending on alignment of figures this could be omitted.

23. Fig.5:
    Please add grid / zero lines.

---

## Referee Report (RR2)

**General Comments:**

The new version of the manuscript has improved and considers all of my comments to the previous version. Aims, procedures and results are now described much clearer. I only have a few minor (mainly technical) comments listed below.

**Specific/technical Comments:**

1. p. 2, l. 19–20:

   "The authors also set the t altitudes limits ..."

   should probably be e.g.:

   "The authors also consider altitudes limits ..."

2. p.4, l. 18:

   "To this application we used two different kind of data sets. The 2-D values ..."

   should be (with ":" instead of "."):

   "To this application we used two different kind of data sets: The 2-D values ..."

3. p. 5, l. 10:

   "refer to as" → "referred to as"

4. p. 8, l. 11 :

   "grows" → "grow"

5. p. 8, l. 14:

   "... this quantity have to be additive ..." → "... this quantity has to be additive ..."

6. p. 8, l. 18–19:

   "We should consider that at any time the pressure value of each level is constant but it does not necessarily happen the same with the geopotential height."

   This sentence is a bit unclear, suggested modification:

   "We should consider that at any time the pressure value of each level is constant but this is not necessarily the case for geopotential height."

7. p. 10, l. 9:

   "Several authors had been reported problems ..." → "Several authors reported problems ..."

---

## Author Response (AR2)

**Answers to the comments on: "Analysis of the geopotential from NWM at GNSS sites and its influence in IWV computation"**

We would like to thank again to the reviewers, especially to reviewer #1, who has contributed significantly to the improvement of this work.

Because both reviewers coincided, we have placed special emphasis on the discussion and conclusions section, including a quantitative assesment of the results and linking them with previous work.

Following, we answer specifically the comments from the reviewers.

**Reviewer #1:**
1. Title
The title was changed according to the suggestion. We didn't use acronyms in the new title.

2. Section 2.2:
We introduced a new notation to clearly distinghish between the geopotential at the pressure levels ($z_i$) and at one of the 4-grid points of the NWM arround the GNSS station ($z_k$).

3. P.4, l. 22-27.
Paragraphs at the 1st and 3rd items were eliminated while the paragraph at the 2nd item was reformulated and moved to section 3.1 (Page 8, line 5).

4. P. 6, under/overestimations of IWV by NWM and 10. Fig 2
According to the suggestion, we modified Figures 2 and 4. A new analysis of the under/overestimations is performed in agreement with the new plots (see Page 6).

5. P. 7, l. 5-6
Corrected. We were talking about the GNSS stations.

6. P. 7, L. 8
The paragraph was re-written. (P. 6, l. 30)

7. P. 7, L 18-20
Corrected

8. P. 8, eqs. 7,8 and related text
Equation 8 is redundant and it was eliminated as well as the correspondent paragraph.

9. P. 10, L. 11-18
In agreement with this suggestion, the section 5: Discussion and conclusions was re-written

**Technical Corrections:**
All the technical corrections were adressed. Here are the details of some particular points:

6. P. 4 L. 14:
This part was removed in the new paragraph.

17. P. 8, L. 27.
 This phrase was eliminated.

21. Figure 3.
The correction was made. We also replace the labels into the plot in agreement with the new notation. The caption was modified too for a sake of clarification.

22. Figure 4.

The figure 4 was modified in agreement with the new figure 2 and adressing the comments of the reviewer

**Reviewer #3:**
General coments:
The Discussion and Conclusions section (Section 5) was re-written including a quantitative appraisement of the results and their comparison with similar findings from previous works.

Specific comments:
P. 1. L. 3.
Yes, microwave radiometers. Corrected.

P2.
Two paragrafs were included in order to take into account the previous strategies of Buehler et al. (2012) and Ning et al. (2013) to deal with the different altitudes when comparing measurements from different data sets.

P. 6. L. 29-35:
Figures 2 and 4 were re-plotted following suggestions from reviewer #1

P. 9. L. 5 and 18:
Corrected

P10. L. 10, 22 and 23-24.
The corrections were addressed. We included comparative percentages and the word "successfully" was eliminated.

Technical corrections:
All corrections suggested by the reviewer were made except those of style (for example: units with super/ sub-indexes in italic fonts).

---

## Author Response (AR3)

**Answers to the comments on: "A numerical method to improve the spatial interpolation of water vapor from numerical weather models: a case study in South and Central America"**

All the specific/technical comments were addressed. Once again, thank a lot to reviewer #1.